# Online Learning and Unlearning: Efficient Algorithms with Near-Optimal Regret Guarantees

## Abstract

We formalize *online learning-unlearning* (OLU) in the Online Convex Optimization (OCO) setting, where a learner updates a model sequentially on a stream of convex losses while accommodating occasional unlearning requests between updates. We require that after a deletion, the distribution of all future outputs is statistically indistinguishable from that of a learner trained on the same stream with the deleted data removed. We propose two OLU algorithms based on Online Gradient Descent (OGD). *Passive OLU* leverages the contractive dynamics of OGD and injects calibrated noise, incurring no additional computation beyond standard OGD; however, its regret depends on the deletion schedule. We then introduce a schedule-robust variant that mitigates this dependence. *Active OLU* employs an offline unlearning algorithm to actively steer the online iterate toward the corresponding retrained trajectory. Under standard convexity and smoothness assumptions, our methods achieve regret comparable to standard OGD, demonstrating that strong online unlearning guarantees can be achieved with minimal loss in learning performance.

## 1 Introduction

Machine unlearning—the process of efficiently removing the influence of specific training data so that a model behaves as if it were retrained without that data—has recently attracted significant attention. Beyond mitigating privacy risks (e.g., membership inference attacks (Shokri et al., 2017; Wang et al., 2023)), unlearning algorithms are central to enforcing regulations such as the GDPR's "right to be forgotten" and can even improve model performance by removing undesirable data points (Goel et al., 2024).

Most prior work on machine unlearning has focused on the offline setting (Neel et al., 2021; Bourtoule et al., 2019; Sekhari et al., 2021; Mu and Klabjan, 2024; Suriyakumar and Wilson, 2022), where data points are unlearned from an already trained model, either all at once or sequentially. However, as these methods operate *after training is complete*, they do not readily extend to settings where data arrives continuously. In practice, data often arrives incrementally, and the model is updated frequently with deletion requests interspersed among these updates. This gives rise to new challenges that offline methods do not address.

We therefore study an *online learning and unlearning* framework. An algorithm receives a stream of examples and produces an iterate at each time step, while also responding to deletion requests. We formalize an online unlearning guarantee (inspired by "delete-to-control" (Cohen et al., 2023)) tailored to this setting: after processing a deletion, the distribution of all subsequent outputs should be statistically indistinguishable from the outputs produced had the deleted example never appeared. We formalize this guarantee in Definition 2.2.

To achieve online unlearning with low overhead, we develop two complementary strategies. *(1) Passive OLU* exploits stability properties of standard online optimization algorithms (e.g., Online Gradient Descent (OGD)) under which the influence of an earlier update decays over time. This enables us to inject calibrated noise at deletion times, with magnitude depending on the index of the unlearned point and the time of unlearning. The resulting regret depends on the deletion schedule; we therefore provide schedule-robust variants that remain effective under adversarial deletion patterns. *(2) Active OLU* leverages an auxiliary offline unlearning algorithm (e.g., ERM-based methods), and actively shifts the online iterate toward the

| Algorithm | Assumption | Regret | Compute |
|---|---|---|---|
| Passive | Strongly Convex | $\log T + \frac{dk^{1.2}}{\varepsilon}$ | 1 |
| | Convex | $\sqrt{\ell T(\log T + k)}$ | 1 |
| Active | Strongly Convex $+$ 4.1 | $\log T + d\sum_{i=1}^{k}\frac{1}{\tau[i]}$ | $\log \tau[i]$ |
| Restart | Strongly Convex | $k\log T$ | 1 |
| | Convex | $k\sqrt{T}$ | 1 |
| Online DP | Strongly Convex | $dk\log^{2.5} T$ | $\log \tau[i]$ |
| | Convex | $k\sqrt{dT\log^{2.5} T}$ | $\log \tau[i]$ |
| Retraining | Strongly Convex | $\log T$ | $\tau[i]$ |
| | Convex | $\sqrt{T}$ | $\tau[i]$ |

Table 1: Summary of results in order terms. For passive unlearning, we report the computation and regret of the deletion-robust version (Algorithm 3) for convex losses. The number of deletions is $k$, and $\ell \leq k$ depends on the deletion schedule. Except for Online DP (which costs $O(\log t)$ at round $t$), all methods have $O(1)$ per-round update cost for non-deletion rounds. 4.1 refers to the Strong Growth Assumption introduced in Section 4.

retrained sequence. This can reduce the noise required (and thus improve regret), at the cost of additional computation and under a mild restriction on the deletion process (Assumption 4.1).

We prove sublinear regret bounds that closely match standard OGD. For convex losses, passive unlearning incurs $O\left(\sqrt{\ell T(\log T + k)}\right)$, where $k$ is the number of deletions and $\ell \leq k$ captures schedule dependence. For $\mu$-strongly convex losses, we obtain schedule-robust guarantees: passive unlearning achieves $O\left(\log T + k^{1.2}\right)$, and active unlearning achieves $O\left(\log T + k\right)$ under Assumption 4.1 with logarithmic computation per deletion. See Table 1 for a detailed comparison of our methods with retraining and online DP baselines.

**Contributions.** Our contributions are three-fold:

- We initiate the *online learning and unlearning* setting and formalize an $(\alpha, \varepsilon)$-online unlearning guarantee (Definition 2.2);

- We design passive and active unlearning algorithms provably satisfying this guarantee;

- We prove that our algorithms add only constant-factor computational overhead while attaining regret that nearly matches the best possible bound achievable without unlearning. A detailed comparison appears in Table 1.

The paper is organized as follows: Section 2 introduces the definitions of online convex optimization, machine unlearning, and online learning-unlearning; Sections 3 and 4 present the passive and active unlearners, with their theoretical guarantees; Section 5 concludes with performance insights.

## 2 Preliminaries and Problem Setup

In Section 2.1, we review online convex optimization (OCO) and formally define unlearning. Then, in Section 2.2, we introduce the *online learning and unlearning* framework.

### 2.1 Preliminaries

**Online Convex Optimization** Let $\mathcal{K}$ denote a convex instance space and let $\mathcal{F}$ be a class of convex cost functions mapping $\mathcal{K}$ to $\mathbb{R}_+$. OCO models an iterative game between a learner and an adversary over $T$

time steps. At each time step $t = 1, \ldots, T$, the learner $\mathcal{A}$ selects a point $z_t \in \mathcal{K}$, while the adversary chooses a convex cost function $f_t \in \mathcal{F}$. The learner then incurs a cost $f_t(z_t)$. Formally, the learner's update rule at each step is

$$z_t = g_t(f_{1:t-1}, z_{1:t-1}), \tag{1}$$

where $g_t$ depends on all previously observed cost functions $f_{1:t-1} = \{f_1, \ldots, f_t\}$ and past outputs $z_{1:t-1} = \{z_1, \ldots, z_{t-1}\}$. The performance of the learner is measured by its *regret*:

$$\mathrm{Regret}_T(\mathcal{A}(f_{1:T})) = \sum_{t=1}^{T} f_t(z_t) - \min_{z \in \mathcal{K}} \sum_{t=1}^{T} f_t(z),$$

where a sublinear regret is desirable. Throughout, we assume an *oblivious* adversary that fixes $f_1, \ldots, f_T$ in advance, and that $\mathcal{K}$ has a bounded diameter $D$. Further assumptions on $f_t$ (e.g. Lipschitzness, smoothness, or strong convexity; see Definition A.1) appear in later sections as needed.

Online Gradient Descent (OGD) is a canonical algorithm for this setting. At time $t$, given a cost function $f_t$ at time $t$, OGD updates its output as:

$$z_t = \Pi_{\mathcal{K}}\left[z_{t-1} - \eta_t \nabla f_{t-1}(z_{t-1})\right], \tag{2}$$

where $\eta_t > 0$ is the learning rate at step $t$ and $\Pi_{\mathcal{K}}$ is the projection operator onto the convex set $\mathcal{K}$. OGD achieves $O(\log T)$ regret for strongly convex and $O(\sqrt{T})$ regret for convex losses (Hazan, 2019; Orabona, 2019).

**Unlearning** Machine unlearning aims to remove *post hoc* the influence of a specific training point on the learned model. Naturally, the gold standard is to retrain from scratch without that point and thus, a good unlearning procedure should produce a model close to the retrained model. Various works use different notions of statistical indistinguishability (Guo et al., 2020; Neel et al., 2021; Chien et al., 2024) to formalize this closeness, inspired by differential privacy (Dwork, 2006). We adopt the following definition of unlearning via Rényi divergence.

**Definition 2.1.** Let $\mathcal{A}$ be a learning algorithm and $\mathcal{R}$ an unlearning algorithm. For a dataset $\mathcal{S}$ and a subset $\mathcal{S}^{\mathcal{U}} \subseteq \mathcal{S}$ of points to be removed, we say that $\mathcal{R}$ is an $(\alpha, \varepsilon)$-unlearner if

$$D_\alpha\left(\mathcal{R}(\mathcal{A}(\mathcal{S}), \mathcal{S}^{\mathcal{U}}) \| \mathcal{R}(\mathcal{A}(\mathcal{S} \setminus \mathcal{S}^{\mathcal{U}}), \emptyset)\right) \leq \varepsilon,$$

where $D_\alpha(\cdot \| \cdot)$ is the $\alpha$-Rényi divergence.

In most unlearning approaches, the unlearning step can be decoupled into a deterministic component which adjusts the current output to approximately match the output that would have been obtained had the algorithm never seen the unlearned point, and a noise component which adds calibrated noise to the adjusted output to obsfucate the approximation error. We define the deterministic unlearning function as $h$ and the perturbation function as $\rho$.

This decomposition provides a useful way to view much of the offline certified unlearning literature. Neel et al. (2021) propose descent-to-delete methods for convex ERM, where $h$ is implemented by running gradient descent on the retained dataset, initialized from the current model, and $\rho$ adds calibrated Gaussian noise to hide the approximation in closeness to retraining. Another line of work leverages local curvature information to steer the model towards the retrained one. For instance, Guo et al. (2020) study certified data removal for regularized linear classifiers, where $h$ is given by a one-step Newton update that approximately removes the contribution of the deleted points, and the remaining residual is hidden through randomized loss perturbation. Similarly, Sekhari et al. (2021) use second-order information computed from the deleted points to approximate the retrained ERM for convex models. Finally, Chien et al. (2024); Koloskova et al. (2025) show that noisy gradient descent can provide certified unlearning guarantees even for ERM with non-convex losses.

## 2.2 Online Learning-Unlearning

We now integrate unlearning into the OCO framework. In an *online learning-unlearning* game, the learner not only submits $z_t \in \mathcal{K}$ each round and incurs $f_t(z_t)$, but may also receive requests to unlearn specific cost functions encountered in the past. Perhaps closest to our work, is the turnstile model in the continual observation literature, which similarly accommodates both insertion and deletion (Jain et al., 2023).

Let $k$ be the number of deletions, and let $\boldsymbol{u} = \{\boldsymbol{u}[1], \ldots, \boldsymbol{u}[k]\}$, $\boldsymbol{\tau} = \{\boldsymbol{\tau}[0], \boldsymbol{\tau}[1], \ldots, \boldsymbol{\tau}[k], \boldsymbol{\tau}[k+1]\}$ denote respectively the *indices* of deleted functions and the *time steps* at which deletions occur. We assume $\boldsymbol{u}[i] \leq \boldsymbol{\tau}[i]$ (a function can only be deleted after it is observed) and $\boldsymbol{\tau}[i-1] \leq \boldsymbol{\tau}[i]$ (deletions arrive in chronological order). For notational convenience, define $\boldsymbol{\tau}[0] = 0$ and $\boldsymbol{\tau}[k+1] = T$.

At time $\boldsymbol{\tau}[i]$, the learner must ensure that the future outputs are indistinguishable from those they would produce if all functions $f_{\boldsymbol{u}[1]}, \ldots, f_{\boldsymbol{u}[i]}$ had never been observed. We call the learner in an online learning-unlearning game an *online learner-unlearner (OLU)*, denoted by $\mathcal{A}_{\mathcal{R}}$.

**Online Learner-Unlearner** An OLU can be constructed from a base online learner. Formally, let $\mathcal{A}$ be a base online learner with update functions $g_1, \ldots, g_T$. An *online learner-unlearner* $\mathcal{A}_{\mathcal{R}}$ implements update function $g_t^{\mathcal{A}_{\mathcal{R}}}$ at time $t$,

$$g_t^{\mathcal{A}_{\mathcal{R}}} = \rho_t \circ h_t \circ g_t, \tag{3}$$

where $h_1, \ldots, h_T$ are the deterministic unlearning functions, and $\rho_1, \ldots, \rho_T$ are the perturbation functions.

For two types of online learning algorithms, OLU is constructed differently from Equation (3). For base learners that explicitly use all past cost functions at each step (e.g. FTL), unlearning procedures ($h_t$ and $\rho_t$) must be applied at every round, essentially treating each update as an independent offline unlearning problem. In contrast, incremental base learners such as OGD update using only the previous iterate and the current loss, e.g. $z_{t+1} = \Pi_{\mathcal{K}}[z_t - \eta_t \nabla f_t(z_t)]$. Between deletion requests, an OLU can therefore follow exactly the same update rule as the base learner. Unlearning is triggered only when a deletion occurs, at which point the algorithm modifies its internal state to remove the influence of the deleted function. However, because past data are encoded in the output of the algorithm, unlearning can be more involved. In this paper, we focus on the latter case.

**Certified OLU** Let $\mathcal{S} = \{f_1, \ldots, f_T\}$ be the cost functions chosen by the adversary, with $\mathcal{S}_t = \{f_1, \ldots, f_t\}$ as the subset up to time $t$. For each $i \in \{0, 1, \ldots, k\}$ and time $t \in [T]$, define the *retained (masked) set* after the first $i$ deletions as

$$\mathcal{S}_t^{(-i)} := \left( f_1^{(-i)}, \ldots, f_t^{(-i)} \right),$$

$$f_s^{(-i)} := \begin{cases} \bot, & s \in \{\mathbf{u}[1], \ldots, \mathbf{u}[i]\}, \\ f_s, & \text{otherwise}, \end{cases} \tag{4}$$

with the convention $\mathcal{S}_t^{(0)} = \mathcal{S}_t$. Here, $\bot$ denotes a skip element: an online learning algorithm does not update its output at time steps where it encounters $\bot$ and ignores $\bot$ in all future updates.

Now define the two output sequences:

$$Z^{(i)} := \mathcal{A}_{\mathcal{R}} \left( \mathcal{S}_{\boldsymbol{\tau}[i+1]-1}, \, \mathbf{u}[1{:}i], \, \boldsymbol{\tau}[1{:}i] \right),$$

$$Z^{(i),-} := \mathcal{A}_{\mathcal{R}} \left( \mathcal{S}_{\boldsymbol{\tau}[i+1]-1}^{(-i)}, \, \mathbf{u}[{:}\,i], \, \boldsymbol{\tau}[1{:}i] \right).$$

Here $Z^{(i)}$ is the output sequence produced when processing the first $i$ deletion requests with online unlearning. In contrast, $Z^{(i),-}$ is the sequence obtained by retraining from scratch on the dataset with the first $i$ deleted functions removed, while additionally injecting noise at each deletion time $\boldsymbol{u}[j]$ ($j \leq i$) with the same distribution as in $Z^{(i)}$. Importantly, the noise scale is determined solely by the perturbation functions $\rho_t$ in Equation (3) using public information (e.g. the indices $\boldsymbol{u}$ and $\boldsymbol{\tau}$), so it does not leak any private information about the unlearned points.

This construction is necessary because the Rényi divergence between a randomized mechanism and a deterministic one is infinite in general. It also mirrors the baseline in offline unlearning (Definition 2.1), where

the comparison is likewise made to a retrained-from-scratch model with matched noise injection. With these definitions, we are ready to present the formal definition of $(\alpha, \varepsilon)$-OLU.

**Definition 2.2.** Fix a deletion schedule $(\boldsymbol{\tau}, \mathbf{u})$ with $1 \leq \mathbf{u}[i] \leq \boldsymbol{\tau}[i] \leq T$ and $\boldsymbol{\tau}[1] < \cdots < \boldsymbol{\tau}[k]$. An online learner–unlearner $\mathcal{A}_\mathcal{R}$ is an $(\alpha, \varepsilon)$-*OLU* if, for all $i = 1, \ldots, k$,

$$D_\alpha \left( Z_{I_i}^{(i)} \| Z_{I_i}^{(i),-} \right) \leq \varepsilon,$$

where $I_i := \{\boldsymbol{\tau}[i], \boldsymbol{\tau}[i] + 1, \ldots, \boldsymbol{\tau}[i+1] - 1\}$, and $Z_{I_i}$ denotes the subsequence $\{Z_t\}_{t \in I_i}$.

In Definition 2.2, we compare the two runs only on the interval $I_i = \{\boldsymbol{\tau}[i], \ldots, \boldsymbol{\tau}[i+1] - 1\}$, i.e., from the $i$th deletion time up to (but not including) the next deletion time. The reason is that the *reference* run changes whenever a new deletion request arrives: after the $(i+1)$st deletion, the appropriate baseline is retraining with one additional loss removed. Thus, the definition enforces a sequence of comparisons, one for each phase between deletions, each using the correct post-deletion dataset.

Although the requirement is stated separately on each interval, it implies persistent protection: once a point is deleted, it remains excluded from the retraining baseline for all subsequent intervals, and therefore remains protected forever.

**OLU regret.** Because the best-in-hindsight comparator changes after each deletion, we define regret in a manner reminiscent of dynamic regret (Zinkevich, 2003; Zhao et al., 2020), allowing the comparator to vary across the intervals between deletions:

$$\underset{T}{\text{Regret}}(\mathcal{A}_\mathcal{R}(\mathcal{S}, \boldsymbol{u}, \boldsymbol{\tau})) := \sum_{i=0}^{k} \sum_{t=\boldsymbol{\tau}[i]+1}^{\boldsymbol{\tau}[i+1]} \left[ f_t(z_t) - f_t(z_i^\star) \right], \tag{5}$$

Recall that we set $\boldsymbol{\tau}[0] := 0$ and $\boldsymbol{\tau}[k+1] := T$. Here $z_i^\star$ is the best fixed decision in hindsight after the first $i$ deletions:

$$z_i^\star \in \arg\min_{z \in \mathcal{K}} \sum_{t=1}^{T} f_t^{(-i)}(z), \tag{6}$$

where $(f_1^{(-i)}, \ldots, f_T^{(-i)})$ is the retained set after the first $i$th deletion (Equation (4)). Allowing $z_i^\star$ to change across the intervals $I_i$ ensures that, within each interval, both the learner and the comparator are evaluated against the same retained history of cost functions, as in the classical online learning setting.

Thus, the changing comparator does not make OLU regret inherently easier than static regret; its effect can go either way, depending on the losses and deletion requests. Nevertheless, the regret guarantees established in later sections hold even when deletions are chosen adversarially, without requiring them to be independent of the loss sequence.

**OLU with DP**

Differential privacy (DP) provides a general route to OLU-type guarantees, as formalized in Proposition 1. However, existing DP-based online learning algorithms, such as Guha Thakurta and Smith (2013); Jain et al. (2012), yield only the approximate OLU guarantee in Definition A.8, a weaker relaxation of the Rényi-divergence-based OLU notion in Definition 2.2. Thus, these algorithms do not directly satisfy our stronger Rényi-style OLU guarantee. In contrast, all algorithms proposed in this paper satisfy this stronger guarantee directly.

To compare regret under the weaker approximate OLU notion, we can convert our Rényi-style OLU guarantee to the corresponding approximate OLU guarantee when $\log(1/\delta) = \Theta(\varepsilon)$ (See Appendix A.1). Then, under the same $(\varepsilon, \delta)$-approximate OLU guarantee, online DP algorithms have regret $O(dk\sqrt{T}/\varepsilon)$ for convex losses and $O(dk \log^{2.5} T/\varepsilon)$ for strongly convex losses, whereas our bounds avoid the multiplicative coupling between $d$ and the $\log T$ factors (see Section 3).

*Proposition* 1. For $k \in \mathbb{N}$, any $(\alpha, \varepsilon)$-RDP online learning algorithm is an $(\alpha/k, k^{1.6}\varepsilon)$-OLU for any deletion set $U$ and deletion-time set $\mathcal{T}$ of size $k$, if $\alpha \geq 2k$.

## 3 Passive Unlearning

---

**Algorithm 1** General Passive OLU

---

**Require:** Sensitivities $\Delta_{1:T}$, cost functions $f_{1:T}$, base updates $g_{1:t}$ of $\mathcal{A}$, learning rates $\eta_{1:T}$, contractive coefficients $\gamma_{1:t}$, deletion time set $\boldsymbol{\tau}$ and deletion index set $\boldsymbol{u}$, privacy parameters $\alpha, \varepsilon$.

1: Initialise $z_1 \in \mathcal{K}$.
2: **for** $t = 2, \ldots, T$ **do**
3:      $z_t \leftarrow g_t(f_{1:t-1}, z_{1:t-1})$. {For OGD, $z_t = \Pi_{\mathcal{K}}[z_{t-1} - \eta_t \nabla f_{t-1}(z_{t-1})]$}
4:      **if** exist $i$ s.t. $t = \tau[i]$ **then**
5:          $\sigma_i = \sqrt{\frac{3i^{1.2}}{\varepsilon}} \prod_{r=\boldsymbol{u}[i]+1}^{\boldsymbol{\tau}[i]} \gamma_r \Delta_{\boldsymbol{u}[i]}$
6:          $z_t \leftarrow g_t(f_{1:t-1}, z_{1:t-1}) + \xi_i, \, \xi_i \sim \mathcal{N}\left(0, \sigma_i^2\right)$
7:      **end if**
8:      **Output** $z_t \leftarrow \Pi_{\mathcal{K}}(z_t)$
9: **end for**

---

In this section, we introduce a general passive online learner-unlearner. This approach incurs no extra computational overhead while ensuring a regret bound that is comparable to a standard online learning algorithm without unlearning. We build upon a base online learner, and illustrate how injecting properly calibrated noise at deletion time steps guarantee unlearning, as formalised below.

**Passive online learner-unlearner**  Our passive online learner-unlearner $\mathcal{A}_{\mathcal{R}}$ (Algorithm 1) is constructed around a base online learning algorithm $\mathcal{A}$ whose update functions are $g_1, \ldots, g_T$. When no deletion request arrives, the algorithm simply follows the updates of $\mathcal{A}$. Upon receiving a deletion request at $\boldsymbol{\tau}[i] \in \boldsymbol{\tau}$, the algorithm first performs the standard update defined by $\mathcal{A}$, then adds calibrated noise $\xi_i \sim \mathcal{N}(0, \sigma_i^2)$ to each coordinate of the output $z_{\boldsymbol{\tau}[i]}$. The noise scale $\sigma_i$ depends on the timing of the deletion request $\boldsymbol{\tau}[i]$, the index of the deleted point $\boldsymbol{u}[i]$, and the sensitivities $\Delta_{\boldsymbol{u}[i]}$, which are considered as public information. Sensitivity is often controlled by the properties of the cost functions, such as Lipschitzness and strong convexity. These properties are inputs to the algorithm and ensure the unlearning guarantee.

Formally, the update function $g_t^{\mathcal{A}_{\mathcal{R}}}$ of Algorithm 1 at time $t$ is defined as $g_t^{\mathcal{A}_{\mathcal{R}}} := \rho_t \circ h_t \circ g_t$ (as defined in Equation (1)), where $h_t(x) = x$ for all $t < T$ is the identity function, indicating no update is performed apart from the base learner's update, and $\rho_t(x)$ injects noise only if $t \in \boldsymbol{\tau}$ is a deletion time:

$$\rho_t(x) = \begin{cases} x & t \notin \boldsymbol{\tau} \\ x + \mathcal{N}(0, \sigma_i^2) & t = \boldsymbol{\tau}[i] \in \boldsymbol{\tau} \end{cases} \tag{7}$$

Algorithm 1 displays a pseudocode of this procedure. In Section 3.1, we introduce three conditions (Condition C1,C2,C3) on the base algorithm's update functions $g_t$ that suffice to establish the online unlearning guarantee in Definition 2.2. Then, in Section 3.2, we derive the regret bound when the base online learner is OGD (Equation (2)), a widely used algorithm satisfying these conditions.

### 3.1 Unlearning guarantee

We first discuss the three sufficient conditions on the update function $g_t$ of the base algorithm $\mathcal{A}$ under which Algorithm 1 is an $(\alpha, \alpha\varepsilon)$-OLU. For any $z_1, z_2, z \in \mathcal{K}$, $z_{1:t-1} \in \mathcal{K}^{t-1}$, any cost function $f \in \mathcal{F}$ and $f_{1:t} \in \mathcal{F}^t$, $\gamma_t \in (0, 1]$ and $\Delta_t < \infty$, the update function $g_t$ satisfies:

$$\textbf{Markovian Output: } g_t(f_{1:t-1}, z_{1:t-1}) = g_t(f_t - 1, z_{t-1}) \tag{C1}$$

$$\gamma\textbf{-Contraction: } \|g_t(f, z_1) - g_t(f, z_2)\|_2 \leq \gamma_t \|z_1 - z_2\|_2 \tag{C2}$$

$$\textbf{Bounded updates: } \|g_t(f, z) - z\|_2 \leq \Delta_t. \tag{C3}$$

Condition C1 ensures that update $g_t$ depends only on the latest cost function and previous output, allowing it to be expressed as $g_t(f_t, z_{t-1})$ and simplifying subsequent conditions. The contraction property in Condition C2 has been central in privacy analyses of DP-SGD (Feldman et al., 2018; Altschuler and Talwar,

Figure 1: Visualization of the output sequence of algorithm $\mathcal{A}$ up to the first deletion $\boldsymbol{\tau}[1]$. The treatment of subsequent deletions follows the same contraction-of-distributions intuition and can be analyzed using PAI.

2022), sampling (Altschuler and Talwar, 2023), and generalisation bounds for SGD (Hardt et al., 2016). Recently, Chien et al. (2024) utilised contraction in designing an *active* unlearning scheme for (noisy) SGD in the offline setting; to the best of our knowledge, our work is the first that applies this idea in an online framework of unlearning. Bounded sensitivity in Condition C3 requires that each single-step update does not lead to arbitrarily large changes in the model. While it is expressed slighly differently from sensitivity notion in the DP literature (Dwork, 2006), it is a standard assumption that holds in many settings, as any continuous function defined on a bounded domain admits a finite sensitivity. Given these conditions, Theorem 3.1 proves that Algorithm 1 is an $(\alpha, \alpha\varepsilon)$-OLU.

**Theorem 3.1.** *Let $\Delta_{1:T} < \infty$ and $\gamma \in (0,1]$. If for all $t \in [T]$, the update function $g_t$ of algorithm $\mathcal{A}$ fulfills Conditions (C1), (C2) and (C3), then Algorithm 1 instantiated with $\mathcal{A}$ is an $(\alpha, \alpha\varepsilon)$-OLU.*

**Proof Sketch.** Consider a single deletion request at time $\boldsymbol{\tau}[1]$ for the point that originally appears at time $\boldsymbol{u}[1]$. Let $\ell = \boldsymbol{\tau}[1] - \boldsymbol{u}[1]$ denote the gap between the point's first inclusion and its requested removal. We compare two output sequences: $(z_t)$, the usual updates by the online learning algorithm $\mathcal{A}$ on all cost functions, and $(z'_t)$, the updates when $f_{\boldsymbol{u}[1]}$ is omitted. As illustrated in Figure 1, for $t < \boldsymbol{u}[1]$, neither process has used $f_{\boldsymbol{u}[1]}$, so $z_t = z'_t$. At $t = \boldsymbol{u}[1]$, $(z_t)$ takes one gradient step on $f_{\boldsymbol{u}[1]}$ while $(z'_t)$ does not, creating a maximum difference of $\Delta_{\boldsymbol{u}[1]}$ (due to C3). For subsequent steps, both follow the same $\gamma$-contractive updates, shrinking their distance by a factor of $\gamma \leq 1$ each time. By $t = \boldsymbol{\tau}[1]$, the distance is at most $\gamma^\ell \Delta_{\boldsymbol{u}[1]}$. Injecting suitably calibrated Gaussian noise of scale proportional to $\gamma^\ell \Delta_{\boldsymbol{u}[1]}$ makes the two processes statistically indistinguishable under Rényi divergence (see Lemma A), yielding the $(\alpha, \alpha\varepsilon)$-OLU guarantee.

For multiple deletions, we must control the divergence between two stochastic processes across several intervention times. One process follows a sequence of time-varying $\gamma_t$-contractive maps and receives Gaussian noise injections at deletion times; the reference process follows the same maps, except that at the (retrained) skip corresponding to a deletion, the effective update can be viewed as an identity step, i.e., a contraction factor 1. To propagate indistinguishability across multiple deletions, we invoke Lemma B.4, a strengthened privacy amplification by iteration (PAI) result. The original PAI theorem is insufficient here because it relies on a single uniform upper bound on contraction over all maps. In our case, this would be 1, thereby obscuring the benefit of genuinely contractive steps. Our lemma instead accommodates time-varying contraction and requires contractiveness for only one of the two sequences. Applying it ensures the indistinguishability guarantee to hold after each deletion. See Appendix B.1 for the complete proof. □

Finally, OGD (Equation 2) satisfies Conditions C1, C2, and C3 under standard smoothness and convexity assumptions. As a result, Algorithm 1 instantiated with OGD is an $(\alpha, \alpha\varepsilon)$-OLU.

**Corollary 3.2.** *Assume each cost function $f_t$ is $\beta$-smooth and convex. Then Algorithm 1 with OGD update step and $\eta \leq 2/\beta$ is an $(\alpha, \alpha\varepsilon)$-OLU for $\gamma = 1$. If the cost functions are $\beta$-smooth and $\mu$-strongly convex, then the same algorithm with $\eta_t = \frac{1}{\mu t}$ is an $(\alpha, \alpha\varepsilon)$-OLU for $\gamma_t = 1 - \frac{1}{t}$.*

### 3.2 Regret guarantee

In this section, we analyze the regret of our passive online unlearning algorithm when OGD is used as the base learner. For strongly convex losses, we obtain a near-optimal regret bound of $O(\log T + k^{1.2})$. For convex losses, we prove a near-optimal regret bound under favorable deletion schedules. Although adversarial schedules can degrade the guarantee, we also propose a simple modification that restores a schedule-robust regret bound.

**Strongly convex loss**  We first establish a regret bound for passive unlearning that requires only constant additional computation at deletion rounds (Theorem 3.3). For favorable deletion schedules (e.g. $k = O((\log T)^{5/6})$ or $\boldsymbol{\tau}[i] > k^{2.2}$) this bound matches the standard $O(\log T)$ scaling of OGD without unlearning.

**Theorem 3.3.** *Suppose $\boldsymbol{u}$ (deletion indices) and $\boldsymbol{\tau}$ (deletion times) are each of size $k$. Assume the cost functions are $L$-Lipschitz, $\beta$-smooth, and $\mu$-strongly convex and $\boldsymbol{u}[i] > \frac{1}{2} + \frac{\beta}{2\mu}$ for $i \in [k]$. Then for all $T \geq k$, the regret of Algorithm 1 with deletions defined by $(\boldsymbol{u}, \boldsymbol{\tau})$ is upper bounded by*

$$\frac{L^2}{\mu}(1 + \log T + 2(k-1)) + \frac{3dL^2}{2\mu\varepsilon} \sum_{i=1}^{k} \frac{i^{1.2}}{\boldsymbol{\tau}[i]}.$$

The last term $\frac{3d}{2\mu\varepsilon} \sum_{i=1}^{k} \frac{i^{1.2}}{\boldsymbol{u}[i]}$ captures the additional error introduced by the unlearning algorithm. Moreover, since the $i$th deletion cannot occur before step $i$ (even when receiving a deletion request at every step), $\boldsymbol{\tau}[i] \geq i$, and therefore $\sum_{i=1}^{k} i^{1.2}/\boldsymbol{\tau}[i] \leq k^{1.2}$. In practice, this bound can be loose. The sum can be much smaller than $k^{1.2}$, especially when most deletions occur late in training.

**Convex losses.**  For general convex losses, we apply regularization and consider $\tilde{f}_t(z) = f_t(z) + \frac{\lambda}{2}\|z\|^2$. In Theorem 3.4, the regret scales as $\tilde{O}(\sqrt{T})$ plus an additional term that depends on the deletion schedule through $\mathcal{G}_2(\boldsymbol{\tau}, \boldsymbol{u})$. Here, we use $\tilde{O}(\cdot)$ to suppress logarithmic factors. Then, we propose an algorithm that combines passive unlearning with periodic restarts, and achieves a schedule-independent regret bound of the order $O\big(\sqrt{\ell T(\log T + 2k)}\big)$ where $\ell < k$ is the number of restarts. We also provide guarantees for adaptive learning rates (AdaGrad), at the cost of treating the gradient norms as public.

**Theorem 3.4.** *Suppose $\boldsymbol{u}$ and $\boldsymbol{\tau}$ are each of size $k$. If $f_1, \ldots, f_T$ are $L$-Lipschitz, $\beta$-smooth convex cost functions. Assume $\boldsymbol{u}[i] > \frac{1}{2} + \frac{\beta}{2\mu}$, then for all $T \geq k$, the regret of Algorithm 1 on regularized loss $\tilde{f}_t(z) = f_t(z) + \frac{\sqrt{\log T + 2k}}{2\sqrt{T}}\|z\|^2$ is upper bounded by*

$$(L^2 + D^2)\sqrt{T(\log T + 2k)} + L^2 dk^{1.7}\mathcal{G}_1\sqrt{\frac{3T}{\log T + 2k}},$$

*where $\mathcal{G}_1(\boldsymbol{\tau}, \boldsymbol{u}) = \sqrt{\sum_{i=1}^{k} \frac{\boldsymbol{\tau}[i]^2}{\boldsymbol{u}[i]^4}}$.*

Unlike the strongly convex case, in the convex setting we have $\gamma = 1$, so the effect of the decreasing learning rate dominates any contractive behavior. As Theorem 3.4 shows, deletions that occur later (larger $\boldsymbol{u}[i]$) lead to smaller regret. In particular, if $\boldsymbol{\tau}[i] = o(\boldsymbol{u}[i]^2/k^2)$, then $\mathcal{G}_1(\boldsymbol{\tau}, \boldsymbol{u})$ is constant in $T$ and $k$, and the regret remains $O(\sqrt{T})$, matching OGD without unlearning. To alleviate this dependence on the deletion schedule, we combine the algorithm with periodic start, which will be introduced in Section 3.2.

The regret bound we obtain by optimizing a regularized loss has an extra $O(\sqrt{\log T})$ factor compared to the optimal regret for OCO without unlearning. This gap is partly an artifact of the proof technique: in standard OCO, optimal rates are typically derived by optimizing the original convex losses directly rather than a regularized surrogate. However, the same technique cannot be applied in our setting, as the comparator is dynamic and the analysis hinges on stability of its trajectory. When the losses are only convex (not strongly convex), directly optimizing the unregularized losses provides too little curvature to control this stability, and the resulting regret bound can blow up.

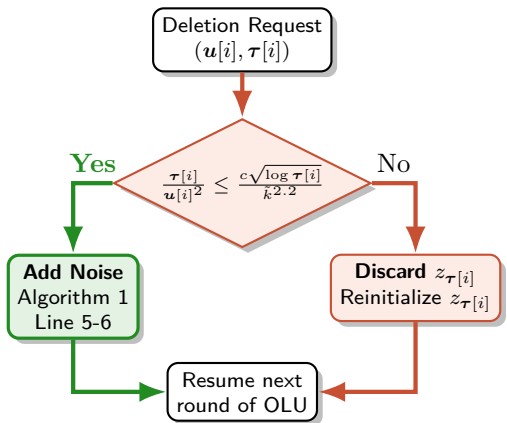

Figure 2: Passive unlearning with periodic restart

In the following, we therefore consider adaptive learning rates (Duchi et al., 2011; Srebro et al., 2012). We show that under reasonable deletion schedules, this lets us eliminate the $O(\sqrt{\log T})$ term, at the cost of making the gradient norms public.

**Theorem 3.5.** *Let $f_1, \ldots, f_T$ be convex, $L$-Lipschitz and $\beta$-smooth and suppose $\boldsymbol{u}$ and $\boldsymbol{\tau}$ are each of size $k$. Define $p(t) = \sum_{i=1}^{t} \|\nabla f_i(z_i)\|_2^2$, if there exists some $u_0 \geq 1$ such that $p(u_0) \geq \beta^2/4$ and $\boldsymbol{u}[i] \geq u_0$, then the expected regret of Algorithm 1 with adaptive learning rate $\eta_t = \frac{D}{\sqrt{p(t)}}$ is*

$$O\left( D^2 \beta + D \sqrt{\sum_{i=0}^{k} \sum_{t=\boldsymbol{\tau}[i]+1}^{\boldsymbol{\tau}[i+1]} f_t(z_i^\star) + \mathcal{G}_2(\boldsymbol{\tau}, \boldsymbol{u}, \mathcal{S})} \right)$$

*where $z_i^\star$ is a best-in-hindsight solution after the $k^{th}$ deletion, and $\mathcal{G}_2(\boldsymbol{\tau}, \boldsymbol{u}, \mathcal{S}) = dk^2 L^2 D^2 \sqrt{\beta \sum_{i=1}^{k} \frac{p(\boldsymbol{\tau}[i])}{p(\boldsymbol{u}[i])^2}}$,*

The passive unlearner with an adaptive learning rate does not explicitly require $\boldsymbol{\tau}[i] = O(\boldsymbol{u}[i])$ and instead accounts for the algorithm's performance over time. Specifically, if the post-deletion gradients do not grow significantly *i.e.* $\mathcal{G}_2(\boldsymbol{\tau}, \boldsymbol{u}, \mathcal{S}) = O(\sqrt{T})$, the additional regret from unlearning stays $O(\sqrt{T})$. The bound also contains a term $D\sqrt{\sum_{i=0}^{k} \sum_{t=\boldsymbol{\tau}[i]+1}^{\boldsymbol{\tau}[i+1]} f_t(z_i^\star)}$, which depends on the best-in-hindsight comparator. In the worst case this term is $O(\sqrt{T})$, but it can be substantially smaller when the best-in-hindsight estimator incurs low loss, yielding a tighter bound (Srebro et al., 2012). Finally, the adaptive method removes the need to know the Lipschitz constant of the cost functions in advance.

**Handling unfavorable deletion schedules**  For convex losses, the regret bound depends on the deletion schedule. As discussed above, an unfavorable schedule can cause the regret to blow up through the schedule-dependent term $\mathcal{G}_1$. To mitigate this, we propose Algorithm 3. Upon receiving an unlearning request for index $\boldsymbol{u}[i]$ at time $\boldsymbol{\tau}[i]$, we check

$$\frac{\boldsymbol{\tau}[i]}{\boldsymbol{u}[i]^2} \leq \frac{c\sqrt{\log \boldsymbol{\tau}[i] + 2\tilde{k}}}{L^2 d\tilde{k}^{2.2}},$$

where $\tilde{k}$ is an upper bound on $k$.

If the condition holds, we perform passive unlearning by adding calibrated noise; otherwise, we discard the current State and restart the algorithm. This strategy yields the following regret guarantee, where $\ell \leq k$ denotes the number of restarts.

---

**Algorithm 2** Active OLU with descent-to-delete unlearner

---

**Require:** Convex set $\mathcal{K} \subseteq \mathbb{R}^d$; costs $f_1, \ldots, f_T$ that are $L$-Lipschitz, $\mu$-strongly convex and $\beta$-smooth; OGD
step sizes $\eta_1, \ldots, \eta_T$; deletion times $\tau[1] < \cdots < \tau[k] \subseteq [T]$ and deletion indices $u[1], \ldots, u[k]$ with
$\tau[i-1] < u[i] \le \tau[i]$; privacy $\varepsilon > 0$; parameter $\omega > 1$; GD step count $I_i$.

1: $\eta_{\text{gd}} \leftarrow \frac{2}{\beta+\mu}, \quad \gamma \leftarrow \frac{\beta-\mu}{\beta+\mu}$.
2: Initialize $z_1 \in \mathcal{K}$.
3: **for** $t = 1, 2, \ldots, T$ **do**
4:       Standard OGD: $\hat{z}_{t+1} \leftarrow \Pi_{\mathcal{K}}[z_t - \eta_t \nabla f_t(z_t)]$
5:       $z_{t+1} \leftarrow \hat{z}_{t+1}$
6:       **if** $t = \tau[i]$ for some $i \in \{1, \ldots, k\}$ **then**
7:           $U_i \leftarrow \{u[1], \ldots, u[i]\}, \quad [t] \leftarrow \{1, \ldots, t\}$
8:           $z \leftarrow \hat{z}_{t+1}$
9:           In the remaining set $\{f_1, \ldots, f_{\tau[i]}\} \setminus \{f_{u[1]}, \ldots, f_{u[i]}\}$, select the function $\tilde{f}$ with the smallest index.
10:          **for** $r = 1$ to $I_i$ **do**
11:              $z \leftarrow \Pi_{\mathcal{K}}[z - \eta_{\text{gd}} \nabla \tilde{f}(z)]$
12:          **end for**
13:          PAI Noise schedule: $\sigma_i^2 \leftarrow \frac{\omega i^\omega \gamma^{2I_i} L^2}{2(\omega-1)\varepsilon}$
14:          Draw $\xi_i \sim \mathcal{N}(0, \sigma_i^2 I_d)$ and set $z_{t+1} \leftarrow z + \xi_i$
15:       **end if**
16:       **output** $z_{t+1}$
17: **end for**

---

**Corollary 3.6.** *Assume the number of restarts is $\ell$. Then under the same assumptions as Theorem 3.4, combining passive unlearning with restarts yields*

$$\mathbb{E}\left[\underset{T}{\text{Regret}}(\mathcal{R}_{\mathcal{A}}(\mathcal{S}, \boldsymbol{u}, \boldsymbol{\tau}))\right] = O\left(\sqrt{\ell T (\log T + 2k)}\right).$$

Note that the bound still depends on the deletion schedule through $\ell$, since $\ell$ counts how often the schedule fails the above criterion and triggers a restart. In the worst case, $\ell$ can be as large as $k$, if every deletion triggers a restart. Nevertheless, this mechanism prevents pathological schedules from having a catastrophic effect on regret: when the criterion is violated, we restart instead of injecting excessively large noise.

## 4 Active Unlearning

The passive approach in Section 3 passively exploits OGD's properties without explicitly moving the current output toward the retrained solution. Analyzing deterministic OGD updates directly for unlearning can be difficult, and to the best of our knowledge, it has not been done before. Instead, we leverage the *descent-to-delete* method of Neel et al. (2021), originally proposed for ERM, and integrate it into an active online learner-unlearner (OLU) (Algorithm 2).

**Active OLU via Descent-to-Delete (Neel et al., 2021)** Of the various ERM-based unlearning algorithms (Sekhari et al., 2021; Neel et al., 2021; Suriyakumar and Wilson, 2022), we adapt the descent-to-delete approach of Neel et al. (2021) thanks to its simplicity, computational efficiency, and certified unlearning guarantee[1]. In essence, this algorithm unlearns a set of points by running a few gradient-descent iterations on the remaining data. Algorithm 2 combines OGD as the base learner with descent-to-delete as the deterministic unlearning function. The procedure alternates between two modes: during regular learning, it follows the OGD update rule. When a request arrives to delete a point $f_{\boldsymbol{u}[i]}$, the algorithm first selects a cost function $f_i$, $i \ne \boldsymbol{u}[i] \le \boldsymbol{\tau}[i-1]$ in the remaining set, and then perform $I_i$ gradient-descent steps on $f_\ell$. Finally, it injects calibrated noise.

---

[1]Our framework also accommodates other ERM unlearning schemes, e.g. Sekhari et al. (2021); Qiao et al. (2025).

Because the delete-to-descent algorithm is tailored to ERM rather than OGD, we impose a strong growth condition (SGC; Assumption 4.1) to control the gap between the OGD iterate and the corresponding ERM solution. As a result, we are allowed to use a single $f_i$ in the remaining set during unlearning. SGC is a standard assumption in convergence analyses of (stochastic) gradient methods (Schmidt and Roux, 2013; Vaswani et al., 2019; Tseng, 1998; Solodov, 1998).

**Assumption 4.1** (Strong Growth Condition (Tseng, 1998; Solodov, 1998))**.** For $j \in [k]$, $F_j = \sum_{i=1}^{T} f_t^{(-j)}$ satisfies strong growth condition if there exists a positive constant $B > 0$ such that for all $z \in \mathcal{Z}$

$$\max_{i \in [T]} \left\| \left( f_i^{(-j)} \right)' (z) \right\| \leq B \left\| F_j'(z) \right\|.$$

In our setting, SGC limits the extent to which any single loss can dominate the online updates and thereby pull the OGD trajectory far from the ERM solution. This control is essential for applying ERM-based unlearning methods to an OGD trajectory. Notably, SGC allows arbitrarily misaligned points as long as they are eventually deleted, but otherwise restricts the adversary to more regular (less adversarial) sequences. Next, in Theorem 4.2, we show that active OLU attains $O(\log T)$ regret, independent of the deletion schedule, with $O(\log \boldsymbol{\tau}[i])$ gradient computations per deletion.

**Theorem 4.2.** *Let $\boldsymbol{u}$ and $\boldsymbol{\tau}$ each have size $k$. For all $i$, assume $\boldsymbol{\tau}[i-1] \leq \boldsymbol{u}[i] \leq \boldsymbol{\tau}[i]$. Suppose each $f_t$ is $L$-Lipschitz, $\mu$-strongly convex, and $\beta$-smooth. Assume Assumption 4.1 holds. For properly selected $I_i$, Algorithm 2 is $(\alpha, \alpha\varepsilon)$-OLU, and for each deletion $i$, the algorithm performs at most $O(\log \boldsymbol{\tau}[i])$ gradient computations, and achieves expected regret*

$$\frac{CL^2 \log T + kD}{2\mu} + \frac{C'dD^2L^2}{\min(\varepsilon, \sqrt{\varepsilon})} \sum_{i=1}^{k} \frac{1}{\boldsymbol{\tau}[i]},$$

*where $C, C'$ are global constants.*

While the regret of passive unlearning matches the order of active OLU, active OLU enjoys a more favorable dependence on the number of deletions. Specifically, the summands in the noise term for active OLU scales like $\frac{1}{\boldsymbol{\tau}[i]}$, whereas those in the passive-unlearning bound scale like $\frac{i^{1.2}}{\boldsymbol{\tau}[i]}$. Consequently, when the number of deletions is large (e.g. $k > \log T$), active OLU is preferable.

## 5 Discussion and Open Problems

In this work, we formalized the problem of Online Learning and Unlearning (OLU) and provided the first certified algorithms capable of handling deletion requests in a truly streaming setting. Table Table 1 summarizes the resulting trade-offs against existing baselines. We next position our results relative to existing approaches and discuss open directions.

**Passive OLU v.s. active OLU.** Our passive OLU attains near-optimal regret for both strongly convex and convex losses without incurring additional computation. The price is that the regret bound can depend on the deletion schedule (cf. Section 3.2), and may degrade under adversarial schedules. In the strongly convex regime, we therefore introduce an active OLU that removes this schedule dependence by performing additional gradient computations at deletion times ($O(\log T)$), under an assumption that restricts the power of the online adversary.

**Comparison with naive baselines.** Two naive baselines are retraining from scratch and discard-and-restart. Retraining attains the best possible regret, but is computationally prohibitive. Discard-and-restart reinitializes OGD after every deletion and trivially achieves $(\alpha, 0)$-OLU, but it leads to a regret scaling linearly with the number of deletion $k$.

**Comparison with differential privacy.** We also compare with the DP-online algorithm (Guha Thakurta and Smith, 2013) as a baseline. However, DP is a global property protecting all points,

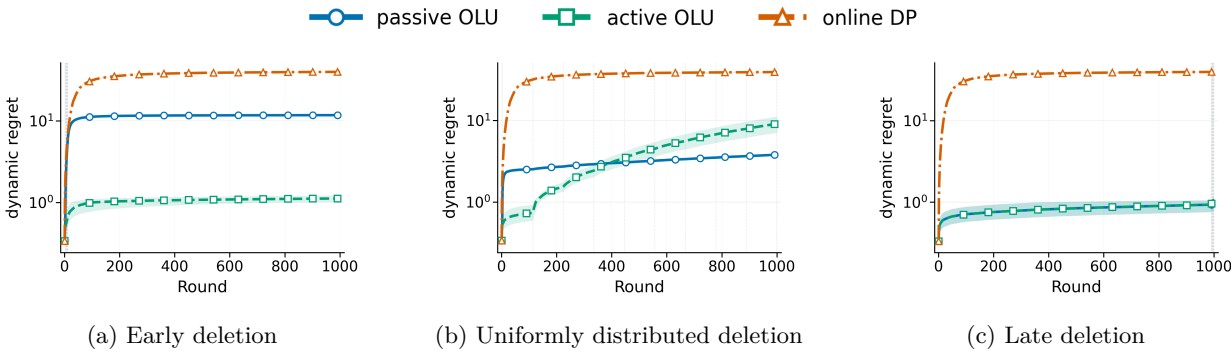

Figure 3: Regret comparison for passive OLU, active OLU, and online DP on quadratic loss functions.

whereas unlearning requires protecting only deleted ones. Consequently, DP baselines incur a regret penalty depending on the dimension $d$ and poly-logarithmic terms in $T$, alongside a computational cost $O(\log t)$ per step.

**Comparison with offline unlearning algorithms.** Compared to offline unlearning algorithms (Chien et al., 2024; Guo et al., 2020; Neel et al., 2021; Sekhari et al., 2021), our method requires weaker assumptions on the loss functions: while most offline approaches rely on strong convexity to establish unlearning guarantees, our passive OLU achieves this under convex losses (Corollary 3.2). However, we cannot fairly compare the accuracy of our algorithm to that of an offline method using online-to-batch conversion, because such conversions average over all past outputs, including those generated before the deleted point was removed, and therefore do not necessarily satisfy the unlearning guarantee.

**Numerical simulations.**

To evaluate passive OLU, active OLU, and online DP, we simulate their performance on strongly convex and smooth quadratic loss functions, defined as $f_t(w) = \frac{1}{2}(x_t^\top w - y_t)^2 + \frac{1}{2}\|w\|_2^2$. The targets are generated as $y_t = x_t^\top w^*$ using randomly sampled inputs $x_t$ and a fixed true parameter $w^*$ drawn from the unit sphere. We restrict active OLU to a maximum of 20 steps. We investigate how deletion schedules impact performance across three scenarios: (1) an adversarial setting for passive OLU featuring initial deletions, (2) a moderate setting with uniformly distributed deletions, and (3) a favorable setting with terminal deletions. Consistent with the comparison of theoretical regret upper bounds shown in Table 1, our results show that active and passive OLU consistently outperform online DP and that their efficiency depends heavily on the deletion schedule.

**Open questions.** A key open direction is to establish regret lower bounds that depend jointly on the horizon $T$ and the number of deletions $k$, thereby clarifying when the accuracy–unlearning trade-offs in Table Table 1 are unavoidable, and in particular whether the dependence on the number of deletions $k$ in our upper bounds is tight. Any such lower bound seemingly requires an explicit model of computational and/or memory limitations: without constraints, an FTL-type procedure could achieve exact unlearning with standard online-learning regret. Related results for exact *offline* unlearning focus instead on space complexity (Ghazi et al., 2023), and do not directly translate to regret lower bounds in the streaming setting.

On the algorithmic side, it is open whether active OLU methods be extended beyond strong convexity or without adversary-restriction assumptions. Additionally, it would be valuable to identify online learners whose update geometry or implicit regularization yields substantially smaller OLU sensitivity (and hence less noise) than OGD, and thus is more suitable for unlearning.

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

# A   Omitted Proofs for Section 2

**Definition A.1.** A function $f : \mathcal{X} \to \mathcal{Y}$, is $L$-Lipschitz if the following hold for all $x, y \in \mathcal{X}$,

$$\|f(x) - f(y)\|_2 \le L \|x - y\|_2.$$

$f$ is called $\mu$-strongly convex if for all $x, y \in \mathcal{X}$,

$$f(x) \ge f(y) + (\nabla f(y))^\top (x - y) + \frac{\mu}{2} \|x - y\|_2^2.$$

$f$ is called $\beta$-smooth if for all $x, y \in \mathcal{X}$,

$$f(x) \le f(y) + (\nabla f(y))^\top (x - y) + \frac{\beta}{2} \|x - y\|_2^2.$$

**Lemma A.2.** *Let $\mathcal{X} \subseteq \mathbb{R}^d$ be convex and let $f : \mathcal{X} \to \mathbb{R}$ be convex. For $\mu > 0$, define $g(x) = f(x) + \frac{\mu}{2}\|x\|_2^2$. Then $g$ is $\mu$-strongly convex with respect to $\|\cdot\|_2$.*

*Proof.* By convexity of $f$, for any $x, y \in \mathcal{X}$ and $\theta \in [0, 1]$,

$$f(\theta x + (1 - \theta)y) \le \theta f(x) + (1 - \theta)f(y).$$

Moreover, the Euclidean norm satisfies the identity

$$\frac{\mu}{2}\|\theta x + (1 - \theta)y\|_2^2 = \frac{\mu}{2}(\theta\|x\|_2^2 + (1 - \theta)\|y\|_2^2 - \theta(1 - \theta)\|x - y\|_2^2).$$

Summing these two, we get

$$g(\theta x + (1 - \theta)y) \le \theta g(x) + (1 - \theta)g(y) - \frac{\mu}{2}\theta(1 - \theta)\|x - y\|_2^2,$$

which is exactly $\mu$-strong convexity of $g$ w.r.t. $\|\cdot\|_2$. $\qquad\qquad\square$

**Definition A.3.** For any two random variables $P, Q$ with corresponding distributions $\mu_P, \mu_Q$ respectively, and for any positive value $\alpha > 0, \alpha \ne 1$, the Rényi divergence between these two distributions is defined as

$$D_\alpha\left(\mu_P \| \mu_Q\right) = \frac{1}{\alpha - 1} \log \int \mu_P(x)^\alpha \mu_Q(x)^{1-\alpha} dx.$$

For simplicity, we sometimes write $D_\alpha\left(P \| Q\right) = D_\alpha\left(\mu_P \| \mu_Q\right)$.

*Proposition* 1. For $k \in \mathbb{N}$, any $(\alpha, \varepsilon)$-RDP online learning algorithm is an $(\alpha/k, k^{1.6}\varepsilon)$-OLU for any deletion set $U$ and deletion-time set $\mathcal{T}$ of size $k$, if $\alpha \ge 2k$.

*Proof.* Let the set of cost functions up to $\boldsymbol{\tau}[i]$, with and without the deleted points indexed by $\mathcal{U}$, be denoted by $\mathcal{S}_i$ and $\mathcal{S}_i'$ respectively, i.e.

$$\mathcal{S}_i = \{f_1, ..., f_{\boldsymbol{\tau}[i]}\}, \quad \mathcal{S}_i' = \{f_1, ..., f_{\boldsymbol{\tau}[i]}\} \setminus \{f_{\boldsymbol{u}[j]}\}_{j=1}^i.$$

Then, for any $i \in \{1, \ldots, k\}$, the number of points that $\mathcal{S}_i$ and $\mathcal{S}_i'$ differ at is upper bounded by $k$. As $\mathcal{A}$ is an online learning algorithm that is $(\alpha, \varepsilon)$-RDP, applying Lemma A.4 on the dataset $\mathcal{S}_i$ and $\mathcal{S}_i'$, we have

$$D_{\frac{\alpha}{k}}(\mathcal{A}(\mathcal{S}_i) \| \mathcal{A}(\mathcal{S}_i')) \le k^{1.6}\varepsilon.$$

**Lemma A.4** (Proposition 2 in Mironov (2017)). *If an algorithm $\mathcal{A}$ is $(\alpha, \varepsilon)$-RDP and if $\alpha \ge 2k$, then for any two dataset $S, S'$ differing by at most $k$ element,*

$$D_{\frac{\alpha}{k}}(\mathcal{A}(S) \| \mathcal{A}(S')) \le k^{1.6}\varepsilon$$

This concludes the proof. $\qquad\qquad\square$

### A.1 Approximate OLU

In this subsection, we first recall two standard privacy notions: approximate differential privacy and Rényi differential privacy.

**Definition A.5** (Approximate differential privacy). Let $\varepsilon \geq 0$ and $\delta \in [0, 1]$. A randomized mechanism $\mathcal{M}$ satisfies $(\varepsilon, \delta)$-differential privacy if, for every pair of neighboring datasets $S, S'$ and every measurable event $E$ in the output space of $\mathcal{M}$,

$$\Pr[\mathcal{M}(S) \in E] \leq e^\varepsilon \Pr[\mathcal{M}(S') \in E] + \delta.$$

The probability is taken over the randomness of $\mathcal{M}$.

**Definition A.6** (Rényi differential privacy). Let $\alpha > 1$ and $\varepsilon_{\mathrm{R}} \geq 0$. A randomized mechanism $\mathcal{M}$ satisfies $(\alpha, \varepsilon_{\mathrm{R}})$-Rényi differential privacy if, for every pair of neighboring datasets $S, S'$,

$$D_\alpha\big(\mathcal{M}(S) \,\|\, \mathcal{M}(S')\big) \leq \varepsilon_{\mathrm{R}},$$

where $D_\alpha(P\|Q)$ denotes the Rényi divergence of order $\alpha$ between distributions $P$ and $Q$, defined as

$$D_\alpha(P\|Q) = \frac{1}{\alpha - 1} \log \mathbb{E}_{X \sim Q}\left[\left(\frac{P(X)}{Q(X)}\right)^\alpha\right].$$

Renyi DP is a stronger notation than approximate DP, and we present Proposition A.7 from Mironov (2017), which shows that every RDP algorithm also satisfies approxiamte DP.

**Proposition A.7** (Proposition 3 from Mironov (2017)). *If a mechanism $f$ satisfies $(\alpha, \varepsilon)$-RDP, then it satisfies*

$$\left(\varepsilon + \frac{\log(1/\delta)}{\alpha - 1}, \delta\right)\text{-}DP$$

*for any $0 < \delta < 1$.*

We then introduce an approximate version of the certified OLU guarantee in Definition A.8, which allows us to compare the regret guarantees of our algorithms with those of existing online algorithms satisfying approximate DP on equal footing.

**Definition A.8** (Approximate OLU). Fix a deletion schedule $(\boldsymbol{\tau}, \mathbf{u})$ with $1 \leq \mathbf{u}[i] \leq \boldsymbol{\tau}[i] \leq T$ and $\boldsymbol{\tau}[1] < \cdots < \boldsymbol{\tau}[k]$. An online learner–unlearner $\mathcal{A}_{\mathcal{R}}$ is an approximate $(\varepsilon, \delta)$-OLU if, for all $i = 1, \ldots, k$, for any output sequence )

$$\Pr(Z_{I_i}^{(i)} \in \mathcal{O}) \leq e^\varepsilon Pr(Z_{I_i}^{(i),-} \in \mathcal{O}) + \delta, \quad \text{and} \quad \Pr(Z_{I_i}^{(i),-} \in \mathcal{O}) \leq e^\varepsilon Pr(Z_{I_i}^{(i)} \in \mathcal{O}) + \delta,$$

where

$$I_i \coloneqq \{\boldsymbol{\tau}[i], \boldsymbol{\tau}[i] + 1, \ldots, \boldsymbol{\tau}[i+1] - 1\},$$

and $Z_{I_i}$ denotes the subsequence $\{Z_t\}_{t \in I_i}$.

Below, we'll show that we can compare the regret of our passive and active OLU algorithm with any algorithm satisfying approximate OLU fairly. By Theorem 3.1, both passive and active OLU satisfy $(\alpha, \alpha\varepsilon_R)$-OLU. Applying the standard RDP-to-approximate-DP conversion (by proposition 3 in [1]) gives that, for every $\delta \in (0, 1)$, the two output sequences from unlearning and retraining satisfy approximate OLU with parameter $\bar{\varepsilon} = \alpha\varepsilon_{\mathrm{R}} + \frac{\log(1/\delta)}{\alpha - 1}$. Optimizing over $\alpha > 1$ gives $\bar{\varepsilon} = \varepsilon_{\mathrm{R}} + 2\sqrt{\varepsilon_{\mathrm{R}} \log(1/\delta)}$. Equivalently, for a target approximate-OLU parameter $\bar{\varepsilon}$, the corresponding Rényi-style parameter in our regret bounds is $\varepsilon_{\mathrm{R}} = \left(\sqrt{\log(1/\delta) + \bar{\varepsilon}} - \sqrt{\log(1/\delta)}\right)^2$. Substituting this value into our regret bounds gives the exact approximate-OLU comparison. In the constant-factor regime where $\log(1/\delta) = \Theta(\varepsilon_{\mathrm{R}})$, for example $\log(1/\delta) = \varepsilon_{\mathrm{R}}$, we have $\bar{\varepsilon} = 3\varepsilon_{\mathrm{R}}$.

Therefore, replacing $\varepsilon_{\mathrm{R}}$ by $\bar{\varepsilon}$ changes only constants, and the regret dependence stated in Theorems 3.3 and 3.4 remains the same order. For other choices of $\delta$, we will state the exact substitution above rather than hiding the dependence.

# B  Omitted Proofs for Section 3

## B.1  Unlearning guarantee of passive unlearning

We denote Euclidean norm by $\|\cdot\|$ or $\|\cdot\|_2$.

**Definition B.1** (shifted Rényi divergence). Let $\mu$, $\nu$ be two distributions. For parameters $e \geq 0$ and $\alpha \geq 1$, the $e$-shifted Rényi divergence between $\mu$ and $\nu$ is defined as

$$D_\alpha^{(e)}(\mu\|\nu) = \inf_{\mu':W_\infty(\mu,\mu')\leq e} D_\alpha(\mu'\|\nu),$$

where $W_\infty$ represents $\infty$-Wasserstein distance.

Shifted Rényi divergence satisfies monotonicity, i.e. For $0 \leq e \leq e'$, $D_\alpha^{(e')}(\mu\|\nu) \leq D_\alpha^{(e)}(\mu\|\nu)$. For a distribution $\zeta$ and a vector $x$, we let $\zeta * x$ denote the distribution of $\eta + x$ where $\eta \sim \zeta$. We define

$$R_\alpha(\zeta,\alpha) = \sup_{x:\|x\|\leq a} D_\alpha(\zeta * x\|\zeta).$$

**Definition B.2** (Contractive Noise Iteration (CNI), Feldman et al. (2018)). Given an initial random state $X_0 \in \mathcal{Z}$, a sequence of contractive functions $\psi_t : \mathcal{Z} \to \mathcal{Z}$, and a sequence of noise distribution $\{\zeta_t\}$, we define the Contractive Noisy Iteration (CNI) by the following update rule:

$$X_{t+1} = \psi_{t+1}(X_t) + \xi_{t+1},$$

where $\xi_{t+1}$ is drawn independently from $\zeta_{t+1}$. We denote the random variable output by this process after $T$ steps as $CNI_T(X_0, \{\psi_t\}, \{\zeta_t\})$.

Next, we provide the proof of the certified unlearning guarantee for passive OLU.

**Theorem 3.1.** *Let $\Delta_{1:T} < \infty$ and $\gamma \in (0,1]$. If for all $t \in [T]$, the update function $g_t$ of algorithm $\mathcal{A}$ fulfills Conditions (C1), (C2) and (C3), then Algorithm 1 instantiated with $\mathcal{A}$ is an $(\alpha, \alpha\varepsilon)$-OLU.*

*Proof.* Let $\mathcal{S} = \{f_1, \ldots, f_T\}$ be the set of cost functions given to the learner over time, $\mathcal{S}^\mathcal{U}$ be the set of deleted points with index in $\mathcal{U} = \{\boldsymbol{u}[1], \ldots, \boldsymbol{u}[k]\}$. Let $\mathcal{T} = \{\boldsymbol{\tau}[1], \ldots, \boldsymbol{\tau}[k]\}$ be the set of deletion times. Let $\mathcal{S}' = \{f_1', \ldots, f_T'\}$ with $f_t' = f_t$ at $t \notin \mathcal{T}$ and $f_t' = \perp$ at $t \in \mathcal{T}$.

We note that the output of Algorithm 1 at time $t$ is a CNI with a sequence of update functions $g_1, \ldots, g_t$, the noise distribution $\zeta_t$ is the Dirac delta distribution at 0 when there is no deletion request, $t \notin \mathcal{T}$, and $\zeta_t = \mathcal{N}(0, \sigma_i^2)$ for the $t = \boldsymbol{\tau}[i]$.

The proof follows an application of Lemma B.3, a more general version of Theorem 22 in Feldman et al. (2018). Compared with Theorem22 in their original paper, Lemma B.3 leverages the fact that the contractive coefficient $\gamma_t$ is not constant for each $t$ and can be strictly smaller than 1, allowing us to achieve the same guarantee in terms of Rényi divergence by adding less noise. Additionally, we note that our theorem only requires $\psi_t$ to be contractive.

**Lemma B.3** (Privacy amplification by iteration with time-varying contraction). *Let $X_T$ and $X_T'$ denote the outputs of $CNI_T(X_0, \{\psi_t\}, \{\zeta_t\})$ and $CNI_T(X_0, \{\psi_t'\}, \{\zeta_t\})$, respectively, where for each $t \in [T]$, $\psi_t$ is contractive with coefficient $\gamma_t \in (0,1]$, i.e. $\psi_t(x) - \psi_t(y)\| \leq \gamma_t\|x - y\| \quad \forall x, y$. Let $s_t := \sup_x \|\psi_t(x) - \psi_t'(x)\|$. Let $a_1, \ldots, a_T$ be a sequence of reals and define the product weights*

$$\Gamma_{i,t} := \prod_{r=i+1}^{t} \gamma_r, \qquad \text{with the convention } \Gamma_{t,t} = 1.$$

*Define the shift sequence $(e_t)_{t=0}^T$ as $e_t := \sum_{i=1}^t \Gamma_{i,t}(s_i - a_i)$. Assume $e_t \geq 0$ for all $t \in [T]$. Then*

$$D_\alpha^{(e_T)}(X_T\|X_T') \leq \sum_{t=1}^T R_\alpha(\zeta_t, a_t).$$

Let $\psi_t(z) = g_t(f_t, z)$ and $\psi'_t(z) = g_t(f'_t, z)$ represents the update function with the $t$th cost functions from $\mathcal{S}$ and $\mathcal{S}'$ respectively. Then,

$$s_t = \sup_{z \in \mathcal{K}} \|g(f_t, z) - g_t(f'_t, z)\| = \sup_{z \in \mathcal{K}} \|\psi_t(z) - \psi'_t(z)\|.$$

As all update functions $g_t$ are $\Delta_t$-bounded (Condition C3), we can compute the value of $s_t$ for all $t \in \{1, \dots, T\}$,

$$s_t = \begin{cases} \Delta_t & t \in \{\boldsymbol{u}[1], \dots, \boldsymbol{u}[k]\} \\ 0 & \text{otherwise} \end{cases}$$

Let $\Gamma_{i,t} = \prod_{r=i+1}^{t} \gamma_r$ be the cumulative contractive coefficient from step $i$ to step $t$. Next, we select the sequence $a_1, \dots, a_T$ such that $R_\alpha(\zeta_t, a_t)$ are bounded and $e_t = \sum_{i=1}^{t} \Gamma_{i,t}(s_i - a_i) \geq 0$ holds for all $t$. By definition of our algorithm (Algorithm 1), the noise is only added at steps $t \in \mathcal{T}$. Therefore, we need to set $a_t = 0$ for all $t \notin \mathcal{T}$ to avoid unbounded $R_\alpha(\zeta_t, a_t)$ when $\zeta_t$ is a Dirac delta distribution at 0. Additionally, for $i \in \{1, \dots, k\}$, we set $a_{\boldsymbol{\tau}[i]} = \Gamma_{i,t}\Delta_{\boldsymbol{u}[i]}$, i.e.

$$a_t = \begin{cases} \Gamma_{\boldsymbol{u}[i], \boldsymbol{\tau}[i]} \Delta_{\boldsymbol{u}[i]} & \text{if } t = \boldsymbol{\tau}[i], i \in \{1, \dots, k\} \\ 0 & \text{otherwise} \end{cases}. \tag{8}$$

This ensures $e_{\boldsymbol{\tau}[i]} = 0$ for all $\boldsymbol{\tau}[i] \in \mathcal{T}$ and $e_t \geq 0$ for all $t$.

For Algorithm 1 to satisfy the unlearning guarantee, it suffices to ensure the following indistinguishability condition holds at time $\boldsymbol{\tau}[i]$. Specifically, for each $i \in \{1, \dots, k\}$, we require that the Rényi divergence between the outputs of the two CNIs at step $\boldsymbol{\tau}[i]$ are bounded by $\varepsilon$ according to Definition 2.2, i.e.,

$$D_\alpha\left(X_{\boldsymbol{\tau}[i]} \| X'_{\boldsymbol{\tau}[i]}\right) \leq \alpha\varepsilon.$$

To prove this, we apply Lemma B.3 with the sequence $a_t$ selected above in Equation (8): for all $\boldsymbol{\tau}[i] \in \mathcal{T}$, where $i \leq k$,

$$D_\alpha\left(X_{\boldsymbol{\tau}[i]} \| X'_{\boldsymbol{\tau}[i]}\right) \leq \sum_{j=1}^{i} R_\alpha(\zeta_{\boldsymbol{\tau}[j]}, a_{\boldsymbol{\tau}[j]})$$

$$\stackrel{(a)}{=} \sum_{j=1}^{i} \frac{\alpha\left(a_{\boldsymbol{\tau}[j]}^2\right)}{2\sigma_j^2} \stackrel{(b)}{=} \sum_{j=1}^{i} \frac{\alpha\varepsilon}{j^\omega} \frac{\omega-1}{\omega} \stackrel{(c)}{\leq} \alpha\varepsilon \tag{9}$$

where step (a) follows from Lemma A, step (b) follows by the definition of $\sigma_j^2 = \frac{j^\omega \omega (\Gamma_{\boldsymbol{u}[i], \boldsymbol{\tau}[i]} \Delta_{\boldsymbol{u}[j]})^2}{2(\omega-1)\varepsilon}$ in our algorithm, and step (c) follows

$$\sum_{j=1}^{i} \frac{1}{j^\omega} = 1 + \sum_{j=2}^{i} \frac{1}{j^\omega} \leq 1 + \int_1^\infty \frac{1}{x^\omega} dx = 1 + \frac{1}{\omega-1} = \frac{\omega}{\omega-1}.$$

*Lemma* A (Corrolary 3 in Mironov (2017)). For any two Gaussian distributions of dimension $d$ with the same variance $\sigma^2 I_d$ but different means $\mu_0, \mu_1$, denoted by $\mathcal{N}\left(\mu_0, \sigma^2 I_d\right)$ and $\mathcal{N}(\mu_1, \sigma^2 I_d)$, the following holds,

$$D_\alpha\left(\mathcal{N}(\mu_0, \sigma^2 I_d) \| \mathcal{N}(\mu_1, \sigma^2 I_d)\right) \leq \frac{\alpha\|\mu_0 - \mu_1\|^2}{2\sigma^2}.$$

Then the same guarantee in Equation (9) extends to the output sequence between $t \in (\boldsymbol{\tau}[i], \boldsymbol{\tau}[i+1])$ by post-processing property of Rényi divergence (Lemma B). Specifically, we consider a $(\boldsymbol{\tau}[i+1]-1-\boldsymbol{\tau}[i])$-dimensional post-processing function $(I, \psi_{\boldsymbol{\tau}[i]+1}, \psi_{\boldsymbol{\tau}[i]+2} \circ \psi_{\boldsymbol{\tau}[i]+1}, \dots, \psi_{\boldsymbol{\tau}[i+1]-1} \circ \dots \psi_{\boldsymbol{\tau}[i]+1})$. Applying Lemma B, we have $D_\alpha\left(X_{\boldsymbol{\tau}[i]:\boldsymbol{\tau}[i+1]-1} \| X'_{\boldsymbol{\tau}[i]:\boldsymbol{\tau}[i+1]-1}\right) \leq \alpha\varepsilon$ as desired.

*Lemma* B (Mironov (2017)). For any Rényi parameter $\alpha \geq 1$, any (possibly random) function $h$, and any two random variable $P, Q$ with corresponding distributions $\mu_P, \mu_Q$,

$$D_\alpha \left( h(P) \| h(Q) \right) \leq D_\alpha \left( P \| Q \right).$$

Finally, note that this analysis doesn't consider projection after noise addition. However, we may additionally project after noise without affecting the privacy guarantee; by post-processing of Rényi divergence, this does not increase divergence, and by non-expansiveness of projection it does not increase the regret bound.

$\square$

*Proof of Lemma B.3.* The proof is by induction and similar to the original proof of Theorem 22 in Feldman et al. (2018).

Let $X_t$ and $X_t'$ denote the t'th iteration of $CNI(X_0, \{\psi_t\}, \{\zeta_t\})$ and $CNI(X_0, \{\psi_t'\}, \{\zeta_t\})$ respectively. For each $t \leq T$, our goal is to show the following equation holds,

$$D_\alpha^{(e_t)} \left( X_t \| X_t' \right) \leq \sum_{i=1}^{t} R_\alpha(\zeta_i, a_i).$$

The base case follows by the definition that $e_0 = 0$ and $X_0 = X_0'$. For the induction step, let $\xi_{t+1}$ denote the random variable drawn from $\zeta_{t+1}$,

$$
\begin{aligned}
D_\alpha^{(e_{t+1})} \left( X_{t+1} \| X_{t+1}' \right) &= D_\alpha^{(e_{t+1})} \left( \psi_{t+1}(X_t) + \xi_{t+1} \| \psi_{t+1}'(X_t') + \xi_{t+1} \right) \\
&\overset{(a)}{\leq} D_\alpha^{(e_{t+1}+a_{t+1})} \left( \psi_{t+1}(X_t) \| \psi_{t+1}(X_t') \right) + R_\alpha(\zeta_{t+1}, a_{t+1}) \\
&\overset{(b)}{\leq} D_\alpha^{(\gamma_{t+1} e_t + s_{t+1})} \left( \psi_{t+1}(X_t) \| \psi_{t+1}(X_t') \right) + R_\alpha(\zeta_{t+1}, a_{t+1}) \\
&\overset{(c)}{\leq} D_\alpha^{(e_t)} \left( X_t \| X_t' \right) + R_\alpha(\zeta_{t+1}, a_{t+1}) \overset{(d)}{\leq} \sum_{i=1}^{t+1} R_\alpha(\zeta_i, a_i),
\end{aligned}
\tag{10}
$$

where step (a) is due to Lemma C, step (b) is due to definition of $e_t = \sum_{i=1}^{t} \Gamma_{i,t} (s_i - a_i)$, i.e.

$$e_{t+1} = \sum_{i=1}^{t+1} \Gamma_{i,t+1}(s_i - a_i) = \gamma_{t+1} \sum_{i=1}^{t} \Gamma_{i,t}(s_i - a_i) + (s_{t+1} - a_{t+1}) = \gamma_{t+1} e_t + s_{t+1} - a_{t+1}.$$

Step (c) follows from Lemma B.4, a modification of privacy amplification of contraction (Lemma 21 in Feldman et al. (2018)), and step (d) follows by the induction hypothesis.

*Lemma* C (Shift-reduction lemma (Feldman et al., 2018)). Let $\mu * \zeta$ denote the distribution of $X + Y$ where $X \sim \mu, Y \sim \zeta$. Let $\mu, \nu$ and $\zeta$ be distributions. Then, for any $\alpha \geq 0$,

$$D_\alpha^{(e)} \left( \mu * \zeta \| \nu * \zeta \right) \leq D_\alpha^{(e+a)} \left( \mu \| \nu \right) + R_\alpha(\zeta, a),$$

where $R_\alpha(\zeta, a) = \sup_{x:\|x\| \leq a} D_\alpha(\zeta * x \| \zeta)$.

**Lemma B.4.** *Suppose $\psi$ is a contractive map with coefficient $\gamma$ and $\sup_x \|\psi(x) - \psi'(x)\| \leq s$. Then, for r.v. $X$ and $X'$,*

$$D_\alpha^{(\gamma e + s)} \left( \psi(X) \| \psi'(X) \right) \leq D_\alpha^{(e)} \left( X \| X' \right).$$

$\square$

*Proof of Lemma B.4.* The proof follows from that of Lemma 21 in Feldman et al. (2018), by removing the step where the contractive coefficient $\gamma$ is upper bounded by 1, and writing shift coefficient explicitly in terms of the contractive coefficient $\gamma$. We repeat their proof here for completeness.

By definition of $D_\alpha^{(e)}(\cdot\|\cdot)$, there exists a joint distribution $(X, Y)$ such that

$$D_\alpha(Y\|X') = D_\alpha^{(e)}(X\|X') \text{ and } \mathbb{P}(\|X - Y\| \leq e) = 1.$$

By the post processing properties of Rényi divergence (Lemma B),

$$D_\alpha(\psi'(Y)\|\psi'(X')) \leq D_\alpha(Y\|X') = D_\alpha^{(e)}(X\|X').$$

Moreover,

$$
\begin{aligned}
\|\psi(X) - \psi'(Y)\| &\leq \|\psi(X) - \psi(Y)\| + \|\psi(Y) - \psi'(Y)\| \\
&\overset{(a)}{\leq} \gamma\|X - Y\| + s \leq \gamma e + s.
\end{aligned}
\tag{11}
$$

where step (a) follows by the definition of contractive maps.

This shows that $(\psi(X), \psi'(Y))$ is a coupling establishing the claimed upper bound on $D_\alpha^{(\gamma e + s)}(\psi(X)\|\psi'(Y))$, and concludes the proof. $\qquad\square$

**Corollary 3.2.** *Assume each cost function $f_t$ is $\beta$-smooth and convex. Then Algorithm 1 with OGD update step and $\eta \leq 2/\beta$ is an $(\alpha, \alpha\varepsilon)$-OLU for $\gamma = 1$. If the cost functions are $\beta$-smooth and $\mu$-strongly convex, then the same algorithm with $\eta_t = \frac{1}{\mu t}$ is an $(\alpha, \alpha\varepsilon)$-OLU for $\gamma_t = 1 - \frac{1}{t}$.*

*Proof of Corollary 3.2.* Our result follows from the contractive properties of projected gradient descent. For standard gradient descent without projection, their contractive parameters are provided in Lemma D.

*Lemma* D (Proposition 18 (Feldman et al., 2018); Lemma 2.2 (Altschuler and Talwar, 2023)). *Gradient descent step is contractive if the loss function is smooth or strongly convex.*

- Suppose a function $\ell : \mathbb{R}^d \to \mathbb{R}$ is convex, twice differentiable, and $M$-smooth. Then the function $\psi$ defined as $\psi(w) = w - \eta\nabla\ell(w)$ is contractive with parameter $(1 - \eta M)$ for $\eta \leq 2/M$.

$$\|\psi(w) - \psi(w')\| \leq \beta\|w - w'\|$$

- Suppose $\ell$ is an $m$-strongly convex and $M$-smooth function for $0 < m \leq M < \infty$. For step size $\eta \leq \frac{2}{M}$, then $\psi$ is contractive with parameter $\max_{\lambda \in \{m, M\}}|1 - \eta\lambda|$.

Since the projection operation is contractive, applying it after each gradient update preserves the contractive nature of gradient descent. Therefore, each step of projected online gradient descent remains contractive with the same coefficient as in the case of gradient descent without projection. This completes the proof. $\quad\square$

## B.2 Regret guarantee of passive unlearning

In this section, we present the proof of regret guarantees for the passive unlearning algorithm.

**Theorem 3.3.** *Suppose $\boldsymbol{u}$ (deletion indices) and $\boldsymbol{\tau}$ (deletion times) are each of size $k$. Assume the cost functions are $L$-Lipschitz, $\beta$-smooth, and $\mu$-strongly convex and $\boldsymbol{u}[i] > \frac{1}{2} + \frac{\beta}{2\mu}$ for $i \in [k]$. Then for all $T \geq k$, the regret of Algorithm 1 with deletions defined by $(\boldsymbol{u}, \boldsymbol{\tau})$ is upper bounded by*

$$\frac{L^2}{\mu}(1 + \log T + 2(k - 1)) + \frac{3dL^2}{2\mu\varepsilon}\sum_{i=1}^{k}\frac{i^{1.2}}{\boldsymbol{\tau}[i]}.$$

*Proof.* Recall that $\boldsymbol{\tau} = \{\boldsymbol{\tau}[1], \ldots, \boldsymbol{\tau}[k]\}$ represents the set of deletion times, and $\boldsymbol{u} = \{\boldsymbol{u}[1], \ldots, \boldsymbol{u}[k]\}$ represents the corresponding index of the points to be deleted. For a set of cost functions $\mathcal{S} = \{f_1, \ldots, f_T\}$, for any $i \in \{1, \ldots, k\}$, let $z_i^\star$ be the best-in-hindsight estimator after the $i$th deletion, i.e.

$$z_i^\star = \arg\min_{z \in \mathcal{K}}\sum_{t=1}^{T}f_t(z) - \sum_{j=1}^{i}f_{\boldsymbol{u}[j]}(z).$$

For the first part of the analysis, we consider the constant best-in-hindsight estimator

$$z^\star = \arg\min_{z \in \mathcal{K}} \sum_{t=1}^{\boldsymbol{\tau}[i]} f_t(z).$$

We first set the learning rate $\eta_t = \frac{1}{\mu t}$. By the updating rule of Algorithm 1, for $t + 1 \notin \mathcal{T}$,

$$
\begin{aligned}
\|z_{t+1} - z^\star\|^2 &= \|\Pi_\mathcal{K}\left[z_t - \eta_t \nabla f_t(z_t) - z^\star\right]\|^2 \\
&\leq \|z_t - z^\star\|^2 + \eta_t^2 \|\nabla f_t(z_t)\|^2 - 2\eta_t(\nabla f_t(z_t))^\top (z_t - z^\star).
\end{aligned}
\tag{12}
$$

If there exists $i$ such that $t + 1 = \boldsymbol{\tau}[i] \in \mathcal{T}$,

$$
\begin{aligned}
\|z_{t+1} - z^\star\|^2 &= \|\Pi_\mathcal{K}\left[z_t - \eta_t \nabla f_t(z_t) + \xi_i - z^\star\right]\|^2 \\
&\leq \|z_t - z^\star + \xi_i\|^2 + \eta_t^2 \|\nabla f_t(z_t)\|^2 - 2\eta_t(\nabla f_t(z_t))^\top (z_t - z^\star + \xi_i).
\end{aligned}
\tag{13}
$$

Rearrange Equation (12) and Equation (13), we have

$$
(\nabla f_t(z_t))^\top (z_t - z_i^\star) \leq
\begin{cases}
\frac{\|z_t - z^\star\|^2 - \|z_{t+1} - z^\star\|^2}{2\eta_t} + \frac{\eta_t \|\nabla f_t(z_t)\|^2}{2} & t + 1 \notin \mathcal{T} \\
-\nabla f_t(z_t)^\top \xi_i + \frac{\|z_t - z^\star + \xi_i\|^2 - \|z_{t+1} - z^\star\|^2}{2\eta_t} + \frac{\eta_t \|\nabla f_t(z_t)\|^2}{2} & t + 1 = \boldsymbol{\tau}[i] \in \mathcal{T}
\end{cases}
\tag{14}
$$

As the loss functions are $\mu$-strongly convexity and by the definition of strong convexity (Definition A.1),

$$f_t(z_t) - f_t(z^\star) \leq (\nabla f_t(z_t))^\top (z_t - z^\star) - \frac{\mu}{2} \|z_t - z^\star\|^2.$$

Summing over $t \in \{1, \dots, T\}$ and substituting Equation (14) into the equation,

$$
\begin{aligned}
\operatorname{Regret}_T(\mathcal{R}_\mathcal{A}(\mathcal{S}, \emptyset, \boldsymbol{\tau})) &= \sum_{t=1}^{T} f_t(z_t) - f_t(z^\star) \\
&\leq \sum_{t=1}^{T} (\nabla f_t(z_t))^\top (z_t - z^\star) - \frac{\mu}{2} \|z_t - z^\star\|^2 \\
&\stackrel{(a)}{=} \underbrace{\sum_{t=1}^{T} \frac{\|z_t - z^\star\|^2 - \|z_{t+1} - z^\star\|^2}{2\eta_t} + \frac{\eta_t \|\nabla f_t(z_t)\|^2}{2} - \frac{\mu}{2} \|z_t - z^\star\|^2}_{A} \\
&\quad + \underbrace{\sum_{i=1}^{k} -(\nabla f_{\boldsymbol{\tau}[i]-1}(z_{\boldsymbol{\tau}[i]-1}))^\top \xi_i + \frac{2\left(z_{\boldsymbol{\tau}[i]-1} - z^\star\right)^\top \xi_i + \|\xi_i\|^2}{2\eta_{\boldsymbol{\tau}[i]-1}}}_{B}
\end{aligned}
$$

where step (a) follows by substituting Equation (14).

This way, we then decompose the expected regret into two parts. In the following, we derive separate upper bounds for each.

$$\mathbb{E}\left[\operatorname{Regret}_T(\mathcal{A}_\mathcal{R}(\mathcal{S}, \emptyset, \boldsymbol{\tau}))\right] = \mathbb{E}_{\xi_{1:k}}[A + B] = \mathbb{E}_{\xi_{1:k}}\mathbb{E}[A|\xi_{1:k}] + \mathbb{E}_{\xi_{1:k}}[B] \tag{15}$$

We first bound the term $\mathbb{E}\left[A|\xi_{1:k}\right]$. Given $\xi_{1:k}$, $z_1, ..., z_T$ are deterministic, we get

$$
\begin{aligned}
\mathbb{E}\left[A|\xi_{1:k}\right] &= \sum_{t=1}^{T} \frac{\|z_t - z^\star\|^2 - \|z_{t+1} - z^\star\|^2}{2\eta_t} + \frac{\eta_t \|\nabla f_t(z_t)\|^2}{2} - \frac{\mu}{2}\|z_t - z^\star\| \\
&= \mathbb{E}\left[\sum_{t=1}^{T}\left(\frac{1}{\eta_t} - \frac{1}{\eta_{t-1}} - \mu\right)\|z_t - z^\star\|^2 + \sum_{t=1}^{T}\frac{\eta_t \|\nabla f_t(z_t)\|^2}{2}\right] \\
&\overset{(a)}{=} \mathbb{E}\left[\sum_{t=1}^{T}\frac{\|\nabla f_t(z_t)\|^2}{2\mu t}\right] \leq \frac{L^2}{2\mu}\left(1 + \log T\right)
\end{aligned}
$$

where step (a) is due to the definition of learning rate $\eta_t$, i.e. $\eta_t = \frac{1}{\mu t}$ and $\frac{1}{\eta_0} = 0$, and step (b) is due to the Lipschitzness of the cost function $f_t$ and that $\sum_{t=1}^{T}\frac{1}{t} \leq 1 + \log T$.

Thus,

$$
\mathbb{E}_{\xi_{1:k}}\left[A\right] = \mathbb{E}_{\xi_{1:k}}\mathbb{E}\left[A|\xi_{1:k}\right] \leq \frac{L^2}{2\mu}(1 + \log T) \tag{16}
$$

It remains to bound $\mathbb{E}_{\xi_{1:k}}\left[B\right]$,

$$
\begin{aligned}
\mathbb{E}_{\xi_{1:k}}\left[B\right] &= \mathbb{E}\left[\sum_{i=1}^{k} -(\nabla f_{\boldsymbol{\tau}[i]-1}(z_{\boldsymbol{\tau}[i]-1}))^\top \xi_i + \frac{2\left(z_{\boldsymbol{\tau}[i]-1} - z_i^\star\right)^\top \xi_i + \|\xi_i\|^2}{2\eta_{\boldsymbol{\tau}[i]-1}}\right] \\
&\overset{(a)}{=} \mathbb{E}\left[\sum_{i=1}^{k}\frac{\|\xi_i\|^2}{2\eta_{\boldsymbol{\tau}[i]-1}}\right] = \sum_{i=1}^{k}\frac{\mathbb{E}\left[\|\xi_i\|^2\right]}{2\eta_{\boldsymbol{\tau}[i]-1}} \\
&\overset{(b)}{=} \sum_{i=1}^{k}\frac{d\sigma_i^2}{2\eta_{\boldsymbol{\tau}[i]-1}} \overset{(c)}{=} \frac{3d}{2\varepsilon\mu}\sum_{i=1}^{k}\frac{i^{1.2}\boldsymbol{\tau}[i]}{\boldsymbol{u}[i]^2}\Gamma_{\boldsymbol{u}[i],\boldsymbol{\tau}[i]}^2
\end{aligned} \tag{17}
$$

where step (a) follows from the fact that $\xi_i$ are independent and have zero mean. Step (b) holds because $\mathbb{E}(\|\xi_i\|^2) = d\sigma_i^2$ for $\xi_i \sim \mathcal{N}(0, \sigma_i^2 I_d)$. Step (c) follows by substituting $\sigma_i^2 = \frac{3i^{1.2}\Gamma_{\boldsymbol{u}[i],\boldsymbol{\tau}[i]}^2}{\varepsilon\mu^2\boldsymbol{u}[i]^2}$.

To simplify the term $\sum_{i=1}^{k}\frac{(\boldsymbol{\tau}[i]-1)}{\boldsymbol{u}[i]^2}\gamma^{2(\boldsymbol{\tau}[i]-\boldsymbol{u}[i])}i^\omega$,

$$
\begin{aligned}
\sum_{i=1}^{k}\frac{i^{1.2}\boldsymbol{\tau}[i]}{\boldsymbol{u}[i]^2}\Gamma_{\boldsymbol{u}[i],\boldsymbol{\tau}[i]}^2 &= \sum_{i=1}^{k}\frac{i^{1.2}\boldsymbol{\tau}[i]}{\boldsymbol{u}[i]^2}\prod_{r=\boldsymbol{u}[i]+1}^{\boldsymbol{\tau}[i]}\left(1 - \frac{1}{r}\right)^2 \\
&\leq \sum_{i=1}^{k}\frac{i^{1.2}\boldsymbol{\tau}[i]}{\boldsymbol{u}[i]^2}\frac{\boldsymbol{u}[i]^2}{\boldsymbol{\tau}[i]^2} \leq \sum_{i=1}^{k}\frac{i^{1.2}}{\boldsymbol{\tau}[i]} \leq \sum_{i=1}^{k}i^{0.2} \leq k^{1.2}.
\end{aligned} \tag{18}
$$

Combining Equations (15) to (18), we have

$$
\mathbb{E}\left[\operatorname*{Regret}_{T}(\mathcal{A}_\mathcal{R}(\mathcal{S}, \emptyset, \boldsymbol{\tau}))\right] \leq \frac{L^2}{\mu}\left(1 + \log T\right) + \frac{3dL^2 k^{1.2}}{2\mu\varepsilon} \tag{19}
$$

We note that up to this point in the proof, we have been computing the regret of our algorithm with respect to a constant best-in-hindsight estimator $z^\star$. However, this differs from our regret definition in Equation (5).

Recall that $\mathcal{S}^\mathcal{U}$ is the set of points deleted by the algorithm with index in $\mathcal{U}$. In the following, we bound the difference between these two regret measures: $\mathbb{E}[\operatorname{Regret}_T(\mathcal{A}_\mathcal{R}(\mathcal{S}, \emptyset, \boldsymbol{\tau}))]$, which corresponds to the regret with a constant comparator, and $\mathbb{E}[\operatorname{Regret}_T(\mathcal{A}_\mathcal{R}(\mathcal{S}, \boldsymbol{u}, \boldsymbol{\tau}))]$, which corresponds to our definition.

$$\left\| \mathbb{E}\left[ \operatorname*{Regret}_{T}(\mathcal{A}_{\mathcal{R}}(\mathcal{S}, \emptyset, \boldsymbol{\tau})) \right] - \mathbb{E}\left[ \operatorname*{Regret}_{T}(\mathcal{A}_{\mathcal{R}}(\mathcal{S}, \boldsymbol{u}, \boldsymbol{\tau})) \right] \right\| = \left\| \sum_{i=1}^{k} \sum_{t=\boldsymbol{\tau}[i-1]}^{\boldsymbol{\tau}[i]} f_t(z^\star) - f_t(z_i^\star) \right\|$$

$$\leq \sum_{i=1}^{k} \sum_{t=\boldsymbol{\tau}[i-1]}^{\boldsymbol{\tau}[i]} L \left\| z^\star - z_i^\star \right\| \tag{20}$$

$$\overset{(a)}{\leq} \frac{2L^2(k-1)}{\mu T} \sum_{i=1}^{k} (\boldsymbol{\tau}[i] - \boldsymbol{\tau}[i-1])$$

$$= \frac{2L^2(k-1)}{\mu},$$

where step (a) follows Lemma B.5 and the fact that $i \leq k$ for all $i \in [k]$.

**Lemma B.5.** *For a set of functions $f_1, ..., f_T$ and for any set of index $\{\boldsymbol{u}[i]\}_{i=1}^{k}$ such that $\boldsymbol{u}[i] \leq T$, let $F(w) = \sum_{i=1}^{T} f_i(w)$ and $F_i(w) = \sum_{i=1}^{T} f_i(w) - \sum_{j=1}^{i-1} f_{\boldsymbol{u}[i]}$. Let $w^\star = \arg\min_w F(w)$ and $w_i^\star = \arg\min_w F_i(w)$. If each $f_i$'s are $\mu$-strongly convex and $L$-Lipschitz, then*

$$\left\| w^\star - w_i^\star \right\| \leq \frac{2(i-1)L}{\mu T}.$$

Combining Equations (19) and (20), and substituting $\omega = 1.2$, we get

$$\mathbb{E}\left[ \operatorname*{Regret}_{T}(\mathcal{A}_{\mathcal{R}}(\mathcal{S}, \boldsymbol{u}, \boldsymbol{\tau})) \right] \leq \frac{L^2}{\mu}(1 + \log T + 2(k-1)) + \frac{3dL^2 k^{1.2}}{2\mu\varepsilon}.$$

This concludes the proof. □

*Proof of Lemma B.5.* For any $i \in \{1, \ldots, k\}$ the definition of $F$,

$$F(w_i^\star) = F_i(w_i^\star) + \sum_{j=1}^{i-1} f_{\boldsymbol{u}[i]}(w_i^\star)$$

$$\leq F_i(w^\star) + \sum_{j=1}^{i-1} f_{\boldsymbol{u}[i]}(w_i^\star) \tag{21}$$

$$= F(w^\star) - \sum_{j=1}^{i-1} f_{\boldsymbol{u}[i]}(w^\star) + \sum_{j=1}^{i-1} f_{\boldsymbol{u}[i]}(w_i^\star)$$

$$\leq F(w^\star) + (i-1)L \left\| w^\star - w_i^\star \right\|,$$

where the last inequality follows by the Lipschitzness of all component functions $f_{\boldsymbol{u}[i]}$.

As each component functions $f_i's$ are $\mu$-strongly convex, the sum of $T$ such functions, $F$, is $T\mu$-strongly convex. Then,

$$F(w_i^\star) \geq F(w^\star) + \frac{T\mu}{2} \left\| w_i^\star - w^\star \right\|^2. \tag{22}$$

Combining Equations (21) and (22) concludes the proof. □

**Theorem 3.4.** *Suppose $\boldsymbol{u}$ and $\boldsymbol{\tau}$ are each of size $k$. If $f_1, \ldots, f_T$ are $L$-Lipschitz, $\beta$-smooth convex cost functions. Assume $\boldsymbol{u}[i] > \frac{1}{2} + \frac{\beta}{2\mu}$, then for all $T \geq k$, the regret of Algorithm 1 on regularized loss $\tilde{f}_t(z) = f_t(z) + \frac{\sqrt{\log T + 2k}}{2\sqrt{T}} \left\| z \right\|^2$ is upper bounded by*

$$(L^2 + D^2)\sqrt{T(\log T + 2k)} + L^2 dk^{1.7} \mathcal{G}_1 \sqrt{\frac{3T}{\log T + 2k}},$$

where $\mathcal{G}_1(\boldsymbol{\tau}, \boldsymbol{u}) = \sqrt{\sum_{i=1}^{k} \frac{\boldsymbol{\tau}[i]^2}{\boldsymbol{u}[i]^4}}$.

*Proof.* For each $f_t$, define the regularized loss as $\tilde{f}_t(z) = f_t(z) + \frac{\lambda}{2} \|z\|^2$. Let $z_i^\star$ be the best-in-hindsight estimator after the $i$th deletion, i.e.

$$z_i^\star = \arg\min_{z \in \mathcal{K}} \sum_{t=1}^{T} f_t(z) - \sum_{j=1}^{i} f_{\boldsymbol{u}[j]}(z).$$

Then, the expected regret is

$$
\begin{aligned}
\mathbb{E}\left[\operatorname*{Regret}_{T}(\mathcal{R}_\mathcal{A}(\mathcal{S}, \mathcal{S}^\mathcal{U}, \mathcal{T}))\right] &= \mathbb{E}\sum_{i=0}^{k} \sum_{t=\boldsymbol{\tau}[i]+1}^{\boldsymbol{\tau}[i+1]} f_t(z_t) - f_t(z_i^\star) \\
&\stackrel{(a)}{=} \mathbb{E}\sum_{i=0}^{k} \sum_{t=\boldsymbol{\tau}[i]+1}^{\boldsymbol{\tau}[i+1]} \left[\tilde{f}_t(z_t) - \tilde{f}_t(z_i^\star) + \frac{\lambda}{2}\left(\|z_i^\star\|^2 - \|z_t\|^2\right)\right] \\
&\stackrel{(b)}{\leq} \mathbb{E}\underbrace{\sum_{i=0}^{k} \sum_{t=\boldsymbol{\tau}[i]+1}^{\boldsymbol{\tau}[i+1]} \left[\tilde{f}_t(z_t) - \tilde{f}_t(z_i^\star)\right]}_{\text{part A}} + TD^2\lambda
\end{aligned}
\tag{23}
$$

where step (a) follows by the definition of $\tilde{f}_t(z) := f_t(z) - \frac{\lambda}{2}\|z\|$ and step (b) follows by the boundedness of the domain $\mathcal{K}$, i.e. for all $z \in \mathcal{K}$, $\|z\|_2 \leq D$.

By Lemma A.2, for all $t \in [T]$, $\tilde{f}_t$ is $\lambda$-strongly convex. Upper bounding part A with Theorem 3.3 yields,

$$\mathbb{E}\left[\text{part A}\right] \leq \frac{L^2}{\lambda}\left(\log T + 2k + \frac{\sqrt{3}dk^{1.7}}{\varepsilon}\mathcal{G}_1\left(\gamma, \mathcal{T}, \mathcal{U}\right)\right) \tag{24}$$

where $\gamma = \frac{\beta/\lambda - 1}{\beta/\lambda + 1}$ and $\mathcal{G}_1\left(\gamma, \mathcal{T}, \mathcal{U}\right) = \sqrt{\sum_{i=1}^{k} \boldsymbol{\tau}[i]^2 \gamma^{4(\boldsymbol{\tau}[i] - \boldsymbol{u}[i])}/\boldsymbol{u}[i]^4}$.

Setting $\lambda = \frac{\sqrt{\log T + 2k}}{\sqrt{T}}$, substituting $\lambda$ into the sum of Equations (23) and (24), we get

$$
\begin{aligned}
\mathbb{E}\left[\operatorname*{Regret}_{T}(\mathcal{R}_\mathcal{A}(\mathcal{S}, \mathcal{S}^\mathcal{U}, \mathcal{T}))\right] &\leq (L^2 + D^2)\sqrt{T(\log T + 2k)} + \frac{\sqrt{3T}L^2 dk^{1.7}\mathcal{G}_1(\gamma, \mathcal{T}, \mathcal{U})}{\sqrt{\log T + 2k}} \\
&\leq (L^2 + D^2)\sqrt{T(\log T + 2k)} + L^2 dk^{1.7}\sqrt{\frac{3T\sum_{i=1}^{k}\frac{\boldsymbol{\tau}[i]}{\boldsymbol{u}[i]^4}}{\log T + 2k}},
\end{aligned}
\tag{25}
$$

where the last inequality uses the fact that $\gamma \leq 1$. This concludes the proof. $\qquad\square$

**Corollary 3.6.** *Assume the number of restarts is $\ell$. Then under the same assumptions as Theorem 3.4, combining passive unlearning with restarts yields*

$$\mathbb{E}\left[\operatorname*{Regret}_{T}(\mathcal{R}_\mathcal{A}(\mathcal{S}, \boldsymbol{u}, \boldsymbol{\tau}))\right] = O\left(\sqrt{\ell T(\log T + 2k)}\right).$$

*Proof.* For some constant $c$ independent of $T, k$ and other parameters,

$$\frac{\boldsymbol{\tau}[i]}{\boldsymbol{u}[i]^2} \leq \frac{c\sqrt{\log \boldsymbol{\tau}[i] + 2k}}{L^2 dk^{2.2}}, \tag{26}$$

Then, the last term in the regret is upper bounded by $c\sqrt{T}$, i.e.

$$L^2 dk^{1.7}\mathcal{G}_2\sqrt{\frac{3T}{\log T + 2k}} \leq c\sqrt{T}.$$

---

**Algorithm 3** General Passive OLU for OLU with restart

---

**Require:** Sensitivities $\Delta_{1:T}$, cost functions $f_{1:T}$, base updates $g_{1:t}$ of $\mathcal{A}$, learning rates $\eta_{1:T}$, contractive coefficient $\gamma$, deletion time set $\boldsymbol{\tau}$, deletion index set $\boldsymbol{u}$, privacy parameters $\varepsilon, \alpha$, and a real $\omega > 1$.

1: Initialise $z_1 \in \mathcal{K}$.
2: **for** Time step $t = 2, \dots, T$ **do**
3: $\quad \mathbf{z_t} = \Pi_{\mathcal{K}} \left[ \mathbf{z_{t-1}} - \eta_t \nabla \mathbf{f_{t-1}}(\mathbf{z_{t-1}}) \right]$
4: $\quad$ **if** there exists $i$ such that $t = \boldsymbol{\tau}[i]$ **then**
5: $\qquad$ **if** $\frac{\boldsymbol{\tau}[i]}{\boldsymbol{u}[i]^2} \leq \frac{c\sqrt{\log \boldsymbol{\tau}[i] + 2k}}{L^2 dk^{2.2}}$ **then**
6: $\qquad\quad \sigma_i \leftarrow \sqrt{\frac{\omega i^\omega}{2(\omega - 1)\varepsilon}} \gamma^{\boldsymbol{\tau}[i] - \boldsymbol{u}[i]} \Delta_{\boldsymbol{u}[i]}$
7: $\qquad\quad z_t \leftarrow z_t + \xi_i, \quad \text{where} \xi_i \sim \mathcal{N}\left(0, \sigma_i^2 \mathcal{I}_d\right)$
8: $\qquad$ **else**
9: $\qquad\quad$ Discard previous outputs, and restart from a random $z_{\boldsymbol{\tau}[i]} \in \mathcal{K}$.
10: $\qquad$ **end if**
11: $\quad$ **end if**
12: $\quad$ **Output** $z_t$
13: **end for**

---

Let $\ell$ be the number of restarts. Let $T_i$ denote the length of the $i$th subgame, and $k_i$ denote the number of deletions in the $i$th subgame. Then, $\sum_i^\ell k_i = k$ and $\sum_{i=1}^\ell T_i = T$.

The regret of this algorithm is

$$
\mathbb{E} \sum_{e=1}^\ell \left( \sum_{i=k_{e-1}+1}^{k_e} \sum_{t=\boldsymbol{\tau}[i]+1}^{\boldsymbol{\tau}[i+1]} \ell_t(z_t) - \sum_{i=k_{e-1}+1}^{k_e} \sum_{t=\boldsymbol{\tau}[i]+1}^{\boldsymbol{\tau}[i+1]} \ell_t(z_i^\star) \right)
$$

$$
\leq \sum_{e=1}^\ell \mathbb{E} \left[ \operatorname*{Regret}_T(\mathcal{R}_\mathcal{A}(\mathcal{S}_{T_e+1:T_{e+1}}, \mathcal{U}_{k_e+1:k_{e+1}}, \mathcal{T})) \right]
$$

$$
= \sum_{e=1}^\ell (L^2 + D^2 + c)\sqrt{(T_e - T_{e-1})(\log(T_e - T_{e-1}) + 2(k_e - k_{e-1}))} \tag{27}
$$

$$
\leq \sqrt{(L^2 + D^2 + c)\ell} \sqrt{\sum_{e=1}^\ell (T_e - T_{e-1})(\log(T_e - T_{e-1}) + 2(k_e - k_{e-1}))}
$$

$$
\leq \sqrt{(L^2 + D^2 + c)\ell} \sqrt{T \log T + 2 \sum_{e=1}^\ell (T_e - T_{e+1})(k_e - k_{e-1})} \leq \sqrt{(L^2 + D^2 + c)\ell T(\log T + 2k)}
$$

$\qquad\qquad\qquad\qquad\qquad\qquad\qquad\qquad\qquad\qquad\qquad\qquad\qquad\qquad\qquad\qquad\qquad\qquad$ $\square$

**Corollary 3.6.** *Assume the number of restarts is $\ell$. Then under the same assumptions as Theorem 3.4, combining passive unlearning with restarts yields*

$$
\mathbb{E} \left[ \operatorname*{Regret}_T(\mathcal{R}_\mathcal{A}(\mathcal{S}, \boldsymbol{u}, \boldsymbol{\tau})) \right] = O\left( \sqrt{\ell T(\log T + 2k)} \right).
$$

*Proof.* For some constant $c$ independent of $T, k$ and other parameters,

$$
\frac{\boldsymbol{\tau}[i]}{\boldsymbol{u}[i]^2} \leq \frac{c\sqrt{\log \boldsymbol{\tau}[i] + 2k}}{L^2 dk^{2.2}}, \tag{28}
$$

Then, the last term in the regret is upper bounded by $c\sqrt{T}$, i.e.

$$
L^2 dk^{1.7} \mathcal{G}_2 \sqrt{\frac{3T}{\log T + 2k}} \leq c\sqrt{T}.
$$

Let $\ell$ be the number of restarts. Let $T_i$ denote the length of the $i$th subgame, and $k_i$ denote the number of deletions in the $i$th subgame. Then, $\sum_i^\ell k_i = k$ and $\sum_{i=1}^\ell T_i = T$.

The regret of this algorithm is

$$
\mathbb{E} \sum_{e=1}^\ell \left( \sum_{i=k_{e-1}+1}^{k_e} \sum_{t=\boldsymbol{\tau}[i]+1}^{\boldsymbol{\tau}[i+1]} \ell_t(z_t) - \sum_{i=k_{e-1}+1}^{k_e} \sum_{t=\boldsymbol{\tau}[i]+1}^{\boldsymbol{\tau}[i+1]} \ell_t(z_i^\star) \right)
$$

$$
\leq \sum_{e=1}^\ell \mathbb{E} \left[ \operatorname*{Regret}_T (\mathcal{R}_\mathcal{A}(\mathcal{S}_{T_e+1:T_{e+1}}, \mathcal{U}_{k_e+1:k_{e+1}}, \mathcal{T})) \right]
$$

$$
= \sum_{e=1}^\ell (L^2 + D^2 + c)\sqrt{(T_e - T_{e-1})(\log(T_e - T_{e-1}) + 2(k_e - k_{e-1}))} \tag{29}
$$

$$
\leq \sqrt{(L^2 + D^2 + c)\ell} \sqrt{\sum_{e=1}^\ell (T_e - T_{e-1})(\log(T_e - T_{e-1}) + 2(k_e - k_{e-1}))}
$$

$$
\leq \sqrt{(L^2 + D^2 + c)\ell} \sqrt{T \log T + 2\sum_{e=1}^\ell (T_e - T_{e+1})(k_e - k_{e-1})} \leq \sqrt{(L^2 + D^2 + c)\ell T(\log T + 2k)}
$$

$\square$

**Theorem 3.5.** *Let $f_1, \ldots, f_T$ be convex, $L$-Lipschitz and $\beta$-smooth and suppose $\boldsymbol{u}$ and $\boldsymbol{\tau}$ are each of size $k$. Define $p(t) = \sum_{i=1}^t \|\nabla f_i(z_i)\|_2^2$, if there exists some $u_0 \geq 1$ such that $p(u_0) \geq \beta^2/4$ and $\boldsymbol{u}[i] \geq u_0$, then the expected regret of Algorithm 1 with adaptive learning rate $\eta_t = \frac{D}{\sqrt{p(t)}}$ is*

$$
O\left( D^2\beta + D\sqrt{\sum_{i=0}^k \sum_{t=\boldsymbol{\tau}[i]+1}^{\boldsymbol{\tau}[i+1]} f_t(z_i^\star) + \mathcal{G}_2(\boldsymbol{\tau}, \boldsymbol{u}, \mathcal{S})} \right)
$$

*where $z_i^\star$ is a best-in-hindsight solution after the $k^{th}$ deletion, and $\mathcal{G}_2(\boldsymbol{\tau}, \boldsymbol{u}, \mathcal{S}) = dk^2 L^2 D^2 \sqrt{\beta \sum_{i=1}^k \frac{p(\boldsymbol{\tau}[i])}{p(\boldsymbol{u}[i])^2}}$,*

*Proof.* For $i \in \{0, 1, ..., k\}$, let

$$
z_i^\star = \arg\min_z \sum_{t=1}^T f_t(z) - \sum_{j=1}^i f_{\boldsymbol{u}[i]}(z)
$$

be the best-in-hindsight estimator after $i$th deletion. Following a similar argument as in Equation (14),

$$
(\nabla f_t(z_t))^\top (z_t - z^\star) \leq \begin{cases} \frac{\|z_t - z^\star\|^2 - \|z_{t+1} - z^\star\|^2}{2\eta_t} + \frac{\eta_t \|\nabla f_t(z_t)\|^2}{2} & t+1 \notin \mathcal{T} \\ -\nabla f_t(z_t)^\top \xi_i + \frac{\|z_t - z^\star + \xi_i\|^2 - \|z_{t+1} - z^\star\|^2}{2\eta_t} + \frac{\eta_t \|\nabla f_t(z_t)\|^2}{2} & t+1 = \boldsymbol{\tau}[i] \in \mathcal{T} \end{cases} \tag{30}
$$

By convexity of the loss function, for each $i$, and its corresponding time steps $t \in [\boldsymbol{\tau}[i-1], \boldsymbol{\tau}[i]]$,

$$
f_t(z_t) - f_t(z_i^\star) \leq \nabla f_t(z_t)^\top (z_t - z_i^\star)
$$

Summing over $t \in \{1, \ldots, T\}$ and substitute in Equation (30), we have

$$
\begin{aligned}
\underset{T}{\text{Regret}}(\mathcal{A}_{\mathcal{R}}(\mathcal{S}, \mathcal{S}^{\mathcal{U}}, \mathcal{T})) &= \sum_{i=0}^{k} \sum_{t=\boldsymbol{\tau}[i]}^{\boldsymbol{\tau}[i+1]} f_t(z_t) - f_t(z_i^{\star}) \\
&\leq \underbrace{\sum_{i=0}^{k} \sum_{t=\boldsymbol{\tau}[i]}^{\boldsymbol{\tau}[i+1]} \frac{\|z_t - z_i^{\star}\|^2 - \|z_{t+1} - z_i^{\star}\|^2}{2\eta_t} + \frac{\eta_t \|\nabla f(z_t)\|^2}{2}}_{A} \\
&\quad + \underbrace{\sum_{i=1}^{k} -(\nabla f_{\boldsymbol{\tau}[i]-1}(z_{\boldsymbol{\tau}[i]-1}))^{\top} \xi_i + \frac{(z_{\boldsymbol{\tau}[i]-1} - z_i^{\star})^{\top} \xi_i + \|\xi_i\|^2}{2\eta_t}}_{B}
\end{aligned}
$$

Then, the expected regret can be upper bounded as

$$
\begin{aligned}
\mathbb{E}\left[\underset{T}{\text{Regret}}(\mathcal{A}_{\mathcal{R}}(\mathcal{S}, \mathcal{S}^{\mathcal{U}}, \mathcal{T}))\right] &= \mathbb{E}_{\xi_{1:k}}[A + B] \\
&= \mathbb{E}_{\xi_{1:k}} \mathbb{E}[A | \xi_{1:k}] + \mathbb{E}_{\xi_{1:k}}[B]
\end{aligned}
\tag{31}
$$

We first bound the first term $\mathbb{E}[A | \xi_{1:k}]$. Given $\xi_{1:k}$, $z_1, ..., z_T$ are deterministic. Thus,

$$
\begin{aligned}
\mathbb{E}[A] = \mathbb{E}[A | \xi_{1:k}] &= \mathbb{E}\left[\sum_{i=0}^{k} \sum_{t=\boldsymbol{\tau}[i]}^{\boldsymbol{\tau}[i+1]} \frac{\|z_t - z_i^{\star}\|^2 - \|z_{t+1} - z_i^{\star}\|^2}{2\eta_t} + \frac{\eta_t \|\nabla f_t(z_t)\|^2}{2}\right] \\
&\leq \sum_{i=0}^{k} \sum_{t=\boldsymbol{\tau}[i]}^{\boldsymbol{\tau}[i+1]} \left(\frac{1}{\eta_t} - \frac{1}{\eta_{t-1}}\right) \frac{\|z_t - z_i^{\star}\|^2}{2} + \sum_{t=1}^{T} \frac{\eta_t \|\nabla f_t(z_t)\|^2}{2} \\
&\leq \frac{D^2}{2\eta_T} + \sum_{t=1}^{T} \frac{D \|\nabla f_t(z_t)\|^2}{2\sqrt{\sum_{j=1}^{t} \|\nabla f_j(z_j)\|^2}} \\
&\overset{(a)}{\leq} \frac{D}{2} \sqrt{\sum_{t=1}^{T} \|\nabla f_t(z_t)\|^2} + D\sqrt{\sum_{t=1}^{T} \|\nabla f_t(z_t)\|^2} = \frac{3D}{2}\sqrt{\sum_{t=1}^{T} \|\nabla f_t(z_t)\|^2}
\end{aligned}
$$

where step (a) follows by Lemma B.6.

**Lemma B.6** (Orabona (2019)). *Let $a_0 \geq 0$ and $f : [0, \infty] \to [0, \infty]$ a nonincreasing function. Then,*

$$
\sum_{t=1}^{T} a_t f\left(a_0 + \sum_{i=1}^{t} a_i\right) \leq \int_{a_0}^{\sum_{t=0}^{T} a_t} f(x) dx.
$$

As the loss functions are $\beta$-smooth, we apply Lemma B.7 to arrive an upper bound on part A.

**Lemma B.7** (Lemma 4.1 in Srebro et al. (2012)). *If $f$ is a $\beta$-smooth function, then the following holds,*

$$
\|\nabla f(x)\|_*^2 \leq 2\beta \left[f(x) - \inf_{y \in \mathbb{R}^d} f(y)\right].
$$

$$
\begin{aligned}
\mathbb{E}[A] = \mathbb{E}[A | \xi_{1:k}] &\leq \frac{3D}{2}\sqrt{\sum_{t=1}^{T} \|\nabla f_t(z_t)\|^2} \\
&\overset{(a)}{\leq} \frac{3D}{2}\sqrt{\beta \sum_{t=1}^{T} \left[f_t(z_t) - \inf_{y \in \mathbb{R}^d} f_t(y)\right]} \overset{(b)}{\leq} \frac{3D}{2}\sqrt{\beta \sum_{t=1}^{T} f_t(z_t)}
\end{aligned}
\tag{32}
$$

where step (a) follows Lemma B.7, and step (b) follows by non-negativity of the loss functions.

It remains to bound $\mathbb{E}_{\xi_{1:k}}[B]$,

$$
\begin{aligned}
\mathbb{E}_{\xi_{1:k}}[B] &= \mathbb{E}\left[\sum_{i=1}^{k} -(\nabla f_{\boldsymbol{\tau}[i]-1}(z_{\boldsymbol{\tau}[i]-1}))^{\top}\xi_i + \frac{\left(z_{\boldsymbol{\tau}[i]-1} - z^{\star}\right)^{\top}\xi_i + \|\xi_i\|^2}{2\eta_{\boldsymbol{\tau}[i]-1}}\right] \\
&\overset{(a)}{=} \mathbb{E}\left[\sum_{i=1}^{k} \frac{\|\xi_i\|^2}{2\eta_{\boldsymbol{\tau}[i]-1}}\right] \overset{(b)}{=} \sum_{i=1}^{k} \frac{di^{\omega}\omega L^2\eta_{\boldsymbol{u}[i]}^2}{4(\omega-1)\varepsilon\eta_{\boldsymbol{\tau}[i]}} \\
&\overset{(c)}{\leq} \frac{dk^{\omega+0.5}\omega L^2}{4\sqrt{3}(\omega-1)\varepsilon}\sqrt{\sum_{i=1}^{k}\frac{\eta_{\boldsymbol{u}[i]}^4}{\eta_{\boldsymbol{\tau}[i]}^2}} \overset{(d)}{=} \frac{dk^{\omega+0.5}\omega L^2 D}{4\sqrt{3}(\omega-1)\varepsilon}\sqrt{\sum_{i=1}^{k}\frac{\sum_{j=1}^{\boldsymbol{\tau}[i]}\|\nabla f_j(z_j)\|^2}{\left(\sum_{j=1}^{\boldsymbol{u}[i]}\|\nabla f_j(z_j)\|^2\right)^2}}
\end{aligned}
\tag{33}
$$

where step (a) follows from the fact that $\xi_i$ are independent and have zero mean. Step (b) follows by $\mathbb{E}\left[\|\xi\|^2\right] = d\sigma_i^2$ for $\xi_i \sim \mathcal{N}(0, \sigma_i^2)$ where $\sigma_i^2 = \frac{i^{\omega}\omega L^2\eta_{\boldsymbol{u}[i]}^2}{2(\omega-1)\varepsilon}$. Step (c) follows by Cauchy-Schwarz inequality and step (d) follows by substituting in $\eta_t = \frac{D}{\sqrt{\sum_{i=1}^{t}\|f_i(z_i)\|^2}}$.

Combining Equations (31) to (33), we have

$$
\sum_{i=0}^{k}\sum_{t=\boldsymbol{\tau}[i]+1}^{\boldsymbol{\tau}[i+1]} f_t(z_t) - f_t(z_i^{\star}) \leq \mathbb{E}[B] + \frac{3D}{2}\sqrt{\beta\sum_{t=1}^{T}f_t(z_t)}
\tag{34}
$$

Then, we apply Lemma B.8 with $x = \sum_{t=1}^{T}f_t(z_t)$ and $a = \frac{9D^2\beta}{4}$, $b = 0$ and $c = \sum_{i=0}^{k}\sum_{t=\boldsymbol{\tau}[i]+1}^{\boldsymbol{\tau}[i+1]}f_t(z_i^{\star}) + \mathbb{E}[B]$

**Lemma B.8.** *Let $a, c > 0$, $b \geq 0$, and $x \geq 0$ such that $x - \sqrt{ax + b} \leq c$. Then $x \leq a + c + 2\sqrt{b + ac}$.*

Therefore, the regret is upper bounded by

$$
\begin{aligned}
\sum_{i=0}^{k}\sum_{t=\boldsymbol{\tau}[i]+1}^{\boldsymbol{\tau}[i+1]} (f_t(z_t) - f_t(z_i^{\star})) &\leq \frac{9D^2\beta}{4} + \mathbb{E}[B] + 3D\sqrt{\beta\sum_{i=0}^{k}\sum_{t=\boldsymbol{\tau}[i]+1}^{\boldsymbol{\tau}[i+1]}f_t(z_i^{\star}) + \mathbb{E}[B]} \\
&\leq \frac{9D^2\beta}{4} + 3D\sqrt{\sum_{i=0}^{k}\sum_{t=\boldsymbol{\tau}[i]+1}^{\boldsymbol{\tau}[i+1]}f_t(z_i^{\star})} \\
&\quad + \frac{dk^{\omega+0.5}\omega L^2 D^2}{4(\omega-1)\varepsilon}\sqrt{\sum_{i=1}^{k}\frac{\beta\sum_{j=1}^{\boldsymbol{\tau}[i]}\|\nabla f_j(z_j)\|^2}{\left(\sum_{j=1}^{\boldsymbol{u}[i]}\|\nabla f_j(z_j)\|^2\right)^2}}.
\end{aligned}
\tag{35}
$$

Taking $\omega = 1.5$ concludes the proof. $\qquad\square$

*Proof of Lemma B.8.* Starting with $x - c \leq \sqrt{ax + b}$, we can square both sides and get

$$
x^2 - (2c + a)x + c^2 \leq b.
$$

Completing the square,

$$
\left(x - \frac{2c + a}{2}\right)^2 \leq b + ac + \frac{a^2}{4}
$$

Taking square root and rearrange terms, we get

$$
\begin{aligned}
x &\le \frac{2c+a}{2} + \sqrt{b + ac + \frac{a^2}{4}} \\
&\le \frac{2c+a}{2} + \sqrt{b+ac} + \sqrt{\frac{a^2}{4}} \\
&= a + c + \sqrt{b+ac}
\end{aligned}
\tag{36}
$$

This concludes the proof. □

## C Omitted Proofs for Section 4

### C.1 Descent-to-delete (Neel et al., 2021)

In this section, we provide the proof of unlearning and convergence guarantees of the active OLU with descent-to-delete as the auxiliary unlearning algorithm (Algorithm 2).

**Theorem 4.2.** *Let $\boldsymbol{u}$ and $\boldsymbol{\tau}$ each have size $k$. For all $i$, assume $\boldsymbol{\tau}[i-1] \le \boldsymbol{u}[i] \le \boldsymbol{\tau}[i]$. Suppose each $f_t$ is $L$-Lipschitz, $\mu$-strongly convex, and $\beta$-smooth. Assume Assumption 4.1 holds. For properly selected $I_i$, Algorithm 2 is $(\alpha, \alpha\varepsilon)$-OLU, and for each deletion $i$, the algorithm performs at most $O(\log \boldsymbol{\tau}[i])$ gradient computations, and achieves expected regret*

$$
\frac{CL^2 \log T + kD}{2\mu} + \frac{C'dD^2L^2}{\min(\varepsilon, \sqrt{\varepsilon})} \sum_{i=1}^{k} \frac{1}{\boldsymbol{\tau}[i]},
$$

*where $C, C'$ are global constants.*

We present the proof of the unlearning guarantee in Appendix C.2 and the proof of the regret guarantee in Appendix C.3. For both proofs, we select $I_i \ge \frac{\omega \log i + 2\log \boldsymbol{\tau}[i] + 2\log L}{2 \log \frac{1}{\gamma}}$.

### C.2 Proof of Unlearning guarantee

*Unlearning guarantee in Theorem 4.2.* Fix $i \in [k]$. Let $S_{\boldsymbol{\tau}[i]} = \{f_1, \ldots, f_{\boldsymbol{\tau}[i]}\}$ and let $S'_{\boldsymbol{\tau}[i]} = \{f'_1, \ldots, f'_{\boldsymbol{\tau}[i]}\}$ be the modified stream where $f'_{\boldsymbol{u}[j]} = \perp$ for all $j \le i$ and $f'_t = f_t$ otherwise.

For each block $j \le i$, let $g_t(z) = \Pi_{\mathcal{K}}(z - \eta_t \nabla f_t(z))$ and $g'_t(z) = \Pi_{\mathcal{K}}(z - \eta_t \nabla f'_t(z))$ be the online updates. Define the block transition maps

$$
\psi_j := g_{\boldsymbol{\tau}[j]} \circ \cdots \circ g_{\boldsymbol{\tau}[j-1]+1}, \qquad \psi'_j := g'_{\boldsymbol{\tau}[j]} \circ \cdots \circ g'_{\boldsymbol{\tau}[j-1]+1}.
$$

Let $\mathcal{U}_{\text{aux}}(\cdot, S, U)$ be the deterministic auxiliary unlearning map (Steps 9–11), i.e. $\mathcal{I}_i$ steps of gradient descent with one of the remaining cost function $f_\ell$, $\ell \notin \boldsymbol{u}$. Then the outputs at deletion time $\boldsymbol{\tau}[i]$ can be written as two CNIs with the same noise $\{\zeta_j\}_{j=1}^{i}$ (Definition B.2):

$$
\text{CNI}_i(z_0, \{\Psi_j\}, \{\zeta_j\}), \qquad \text{CNI}_i(z_0, \{\Psi'_j\}, \{\zeta_j\}),
$$

where $\Psi_j(z) = \mathcal{U}_{\text{aux}}(\psi_j(z), \boldsymbol{\tau}[:j], \boldsymbol{u}[:j])$ and $\Psi'_j(z) = \mathcal{U}_{\text{aux}}(\psi'_j(z), \boldsymbol{\tau}[:j] \setminus \boldsymbol{u}[:j], \emptyset)$. We will apply Lemma B.3.

First, we'll bound the per-stage sensitivity $s_j$. Define

$$
\begin{aligned}
s_j &:= \max_{z \in \mathcal{K}} \|\Psi_j(z) - \Psi'_j(z)\|_2 \\
&= \max_{z \in \mathcal{K}} \left\{ \mathcal{U}_{\text{aux}}(\psi_j(z), \boldsymbol{\tau}[:j], \boldsymbol{u}[:j]) - \mathcal{U}_{\text{aux}}(\psi'_j(z), \boldsymbol{\tau}[:j] \setminus \boldsymbol{u}[:j], \emptyset) \right\}
\end{aligned}
\tag{37}
$$

By adding and subtracting $\mathcal{U}_{\text{aux}}(\psi_j(z), \boldsymbol{\tau}[:j] \setminus \boldsymbol{u}[:j], \emptyset)$ and using triangle inequality,

$$
s_j \le A_j + B_j
\tag{38}
$$

where

$$A_j := \max_{z \in \mathcal{K}} \left\| \mathcal{U}_{\mathrm{aux}}(\psi_j(z), \boldsymbol{\tau}[:j], \boldsymbol{u}[:j]) - \mathcal{U}_{\mathrm{aux}}(\psi_j(z), \boldsymbol{\tau}[:j] \setminus \boldsymbol{u}[:j], \emptyset) \right\|_2$$

is the *effect of the unlearning algorithm starting from the same initial point*, and

$$B_j := \max_{z \in \mathcal{K}} \left\| \mathcal{U}_{\mathrm{aux}}(\psi_j(z), \boldsymbol{\tau}[:j] \setminus \boldsymbol{u}[:j], \emptyset) - \mathcal{U}_{\mathrm{aux}}(\psi_j'(z), \boldsymbol{\tau}[:j] \setminus \boldsymbol{u}[:j], \emptyset) \right\|_2$$

is the *initialization-shift effect*.

*(i) Bound $A_j$ with the definition of the algorithm.*

We note that both $\mathcal{U}_{\mathrm{aux}}(\psi_j(z), \boldsymbol{\tau}[:j], \boldsymbol{u}[:j])$ and $\mathcal{U}_{\mathrm{aux}}(\psi_j(z), \boldsymbol{\tau}[:j] \setminus \boldsymbol{u}[:j], \emptyset)$ represent algorithms that start from the initial point $\psi_j(z)$ and apply $I_i$ steps of gradient descent on the same cost function $f_\ell$, $\ell \in \boldsymbol{\tau}[:j] \setminus \boldsymbol{u}[:j]$. Thus, these two terms are equivalent, i.e.,

$$A_j = 0$$

*(ii) Bound $B_j$ using contraction in $\mathcal{U}_{\mathrm{aux}}$ and in $\psi_j$.*

By Lemma D, as $\mathcal{U}_{\mathrm{aux}}$ applies gradient descent on a $\mu$-strongly convex and $\beta$-smooth function $f_\ell$, each gradient step is $\gamma$ contractive, where $\gamma = \frac{\beta - \mu}{\beta + \mu}$. Thus, the $I_i$ of these gradient descent descent step on $f_\ell$ is $\gamma^{I_i}$-contractive. That is,

$$B_j \leq \gamma^{I_i} \cdot \max_{z \in \mathcal{K}} \|\psi_j(z) - \psi_j'(z)\|.$$

The maps $\psi_j$ and $\psi_j'$ differ only at the single index $\boldsymbol{u}[j] \in (\boldsymbol{\tau}[j-1], \boldsymbol{\tau}[j]]$, so the discrepancy introduced at that step is at most $\eta_{\boldsymbol{u}[j]} L$, as each online gradient descent step with step size $\eta = \frac{1}{\mu t}$ is $1 - \frac{1}{t} \leq 1$-contractive:

$$\max_{z \in \mathcal{K}} \|\psi_j(z) - \psi_j'(z)\| \leq \eta_{\boldsymbol{u}[j]} L$$

Therefore,

$$B_j \leq \gamma^{I_j} \eta_{\boldsymbol{u}[j]} L \leq \gamma^{I_j} L.$$

Combining these terms with Equation (38),

$$s_j \leq \gamma^{I_j} L$$

We are ready to apply PAI. Set $a_j = s_j \leq \gamma^{I_j} L$, so $e_t = \sum_{r=1}^{t} \gamma^{t-r}(s_r - a_r) = 0$ for all $t$.

Then by Lemma B.3,

$$D_\alpha\left(z_{\boldsymbol{\tau}[i]} \| z_{\boldsymbol{\tau}[i]}'\right) \leq \sum_{j=1}^{i} R_\alpha(\zeta_j, a_j) = \sum_{j=1}^{i} \frac{\alpha a_j^2}{2\sigma_j^2} \overset{(a)}{\leq} \sum_{j=1}^{i} \frac{\alpha \varepsilon(\omega - 1)}{\omega j^\omega} \overset{(b)}{\leq} \alpha \varepsilon,$$

where step (a) follows by the definition of $\zeta_j \overset{d}{=} \mathcal{N}(0, \sigma_j^2 I)$ with $\sigma_j^2 = \frac{\omega j^\omega \gamma^{2I_j} L^2}{2(\omega-1)\varepsilon} \geq \frac{\omega j^\omega a_j^2}{2(\omega-1)\varepsilon}$. Step (b) uses $\sum_{j \geq 1} j^{-\omega} \leq \frac{\omega}{\omega - 1}$ for $\omega > 1$. Choosing $\omega = 1.2$ concludes the proof.

Similar to the proof of unlearning guarantee for passive OLU, we may additionally project after noise without affecting the privacy guarantee; by post-processing of Rényi divergence, this does not increase divergence, and by non-expansiveness of projection it does not increase the regret bound. □

## C.3 Proof of regret guarantee

*Regret guarantee in Theorem 4.2 .* Let $\boldsymbol{\tau} = \{\boldsymbol{\tau}[0], \boldsymbol{\tau}[1], \ldots, \boldsymbol{\tau}[k], \boldsymbol{\tau}[k+1]\}$ with $\boldsymbol{\tau}[0] = 0, \boldsymbol{\tau}[k+1] = T$. Define the OGD proposal

$$\tilde{z}_{t+1} := \Pi_{\mathcal{K}}\big(z_t - \eta_t \nabla f_t(z_t)\big).$$

If $t + 1 \neq \boldsymbol{\tau}[i]$ for $i \in \{1, ..., k\}$, then $z_{t+1} = \tilde{z}_{t+1}$; otherwise, then

$$z_{t+1} = z_{\boldsymbol{\tau}[i]} = \mathcal{U}_{\text{aux}}(\tilde{z}_{\boldsymbol{\tau}[i]}, f_{1:\boldsymbol{\tau}[i]}, f_{\boldsymbol{u}[1]:\boldsymbol{u}[i]}) + \xi_i.$$

Recall the best-in-hindsight comparator after the ith deletion is defined in Equation (6) as

$$z_i^\star := \arg\min_{z \in \mathcal{K}} \left( \sum_{t=1}^T f_t(z) - \sum_{j=1}^i f_{\boldsymbol{u}[j]}(z) \right), \quad z_{i,\boldsymbol{\tau}[i]}^\star := \arg\min_{z \in \mathcal{K}} \left( \sum_{t=1}^{\boldsymbol{\tau}[i]} f_t(z) - \sum_{j=1}^i f_{\boldsymbol{u}[j]}(z) \right).$$

We will bound the expected regret defind in Equation (5), namely

$$\mathbb{E}\left[ \sum_{i=0}^k \sum_{t=\boldsymbol{\tau}[i]+1}^{\boldsymbol{\tau}[i+1]} \left( f_t(z_t) - f_t(z_i^\star) \right) \right].$$

First, we'll bound the regret during non-deletion rounds with standard OGD inequality. Fix $i \in \{0, ..., k\}$, and for all $t \in \boldsymbol{\tau}[i], ..., \boldsymbol{\tau}[i+1] - 2$, by $\mu$-strong convexity of $f_t$,

$$f_t(z_t) - f_t(z_i^\star) \leq \nabla f_t(z_t)^\top (z_t - z_i^\star) - \frac{\mu}{2} \|z_t - z_i^\star\|^2. \tag{39}$$

Using the standard OGD inequality as in the proof of Theorem 3.3,

$$\nabla f_t(z_t)^\top (z_t - z_i^\star) \leq \frac{\|z_t - z_i^\star\|^2 - \|z_{t+1} - z_i^\star\|^2}{2\eta_t} + \frac{\eta_t}{2} \|\nabla f_t(z_t)\|^2. \tag{40}$$

Combining Equations (39) and (40), for all such $t$,

$$f_t(z_t) - f_t(z_{i,0}^\star) \leq \frac{\|z_t - z_i^\star\|^2 - \|z_{t+1} - z_i^\star\|^2}{2\eta_t} + \frac{\eta_t}{2} \|\nabla f_t(z_t)\|^2 - \frac{\mu}{2} \|z_t - z_i^\star\|^2. \tag{41}$$

Thus, for $\eta_t = \frac{1}{\mu t}$,

$$\sum_{t=\boldsymbol{\tau}[i]}^{\boldsymbol{\tau}[i+1]-2} f_t(z_t) - f_t(z_{i,0}^\star) \leq \sum_{t=\boldsymbol{\tau}[i]}^{\boldsymbol{\tau}[i+1]-2} \frac{\|z_t - z_i^\star\|^2 - \|z_{t+1} - z_i^\star\|^2}{2\eta_t} + \frac{\eta_t}{2} \|\nabla f_t(z_t)\|^2 - \frac{\mu}{2} \|z_t - z_i^\star\|^2.$$

$$= \frac{\mu(\boldsymbol{\tau}[i]-1)}{2} \|z_{\boldsymbol{\tau}[i]} - z_i^\star\|^2 - \frac{\mu(\boldsymbol{\tau}[i+1]-2)}{2} \|z_{\boldsymbol{\tau}[i+1]-1} - z_i^\star\|^2 + \sum_{t=\boldsymbol{\tau}[i]}^{\boldsymbol{\tau}[i+1]-2} \frac{\eta_t}{2} \|\nabla f_t(z_t)\|^2$$

$$\leq \frac{\mu(\boldsymbol{\tau}[i]-1)}{2} \|z_{\boldsymbol{\tau}[i]} - z_i^\star\|^2 - \frac{\mu(\boldsymbol{\tau}[i+1]-2)}{2} \|z_{\boldsymbol{\tau}[i]-1} - z_i^\star\|^2 + \frac{L^2}{2\mu} \log \frac{\boldsymbol{\tau}[i+1]}{\boldsymbol{\tau}[i]} \tag{42}$$

where the last inequality follows by

$$\sum_{t=\boldsymbol{\tau}[i]}^{\boldsymbol{\tau}[i+1]-2} \frac{\eta_t}{2} \|\nabla f_t(z_t)\|^2 \leq \frac{L^2}{2\mu} \sum_{t=\boldsymbol{\tau}[i]}^{\boldsymbol{\tau}[i+1]-2} \frac{1}{t} = \frac{L^2}{2\mu} \log \frac{\boldsymbol{\tau}[i+1]}{\boldsymbol{\tau}[i]}$$

Substitute Equation (42) into the regret bound, we get

$$\sum_{i=0}^{k}\sum_{t=\boldsymbol{\tau}[i]+1}^{\boldsymbol{\tau}[i+1]}\left(f_t(z_t) - f_t(z_i^\star)\right) = \sum_{i=0}^{k}\sum_{t=\boldsymbol{\tau}[i]}^{\boldsymbol{\tau}[i+1]-2} f_t(z_t) - f_t(z_{i,0}^\star) + \sum_{i=1}^{k+1} f_{\boldsymbol{\tau}[i]-1}(z_t) - f_{\boldsymbol{\tau}[i]-1}(z_i^\star)$$

$$\leq \sum_{i=0}^{k}\frac{L^2}{2\mu}\log\frac{\boldsymbol{\tau}[i+1]}{\boldsymbol{\tau}[i]} + \sum_{i=0}^{k}\frac{\mu(\boldsymbol{\tau}[i]-1)}{2}\|z_{\boldsymbol{\tau}[i]} - z_i^\star\|^2 - \frac{\mu(\boldsymbol{\tau}[i+1]-2)}{2}\|z_{\boldsymbol{\tau}[i]-1} - z_i^\star\|^2$$

$$+ \sum_{i=1}^{k+1} f_{\boldsymbol{\tau}[i]-1}(z_t) - f_{\boldsymbol{\tau}[i+1]-1}(z_i^\star)$$

$$\leq \frac{L^2(\log T + 1) + kD^2}{2\mu} + \underbrace{\sum_{i=0}^{k}\frac{\mu(\boldsymbol{\tau}[i]-1)}{2}\|z_{\boldsymbol{\tau}[i]} - z_i^\star\|^2 - \frac{\mu(\boldsymbol{\tau}[i+1]-1)}{2}\|z_{\boldsymbol{\tau}[i+1]-1} - z_i^\star\|^2}_{\text{jump}}$$

$$+ \underbrace{\sum_{i=1}^{k+1} f_{\boldsymbol{\tau}[i]-1}(z_t) - f_{\boldsymbol{\tau}[i]-1}(z_i^\star)}_{\text{deletion cost}}$$

$$\tag{43}$$

We bound the jump term and the deletion cost term (by Lipschitzness of the cost function),

$$\text{jump} + \text{deletion cost} \leq \frac{\mu(\boldsymbol{\tau}[0]-1)}{2}\left\|z_{\boldsymbol{\tau}[0]} - z_0^\star\right\|^2 + \sum_{i=1}^{k}\frac{\mu(\boldsymbol{\tau}[i]-1)}{2}\left(\left\|z_{\boldsymbol{\tau}[i]} - z_i^\star\right\|^2 - \left\|z_{\boldsymbol{\tau}[i]-1} - z_{i-1}^\star\right\|^2\right) + L\|z_{\boldsymbol{\tau}[i]} - z_i^\star\|$$

$$\leq \sum_{i=1}^{k}\frac{\mu(\boldsymbol{\tau}[i]-1)}{2}\left(\left\|z_{\boldsymbol{\tau}[i]} - z_i^\star\right\|^2 - \left\|z_{\boldsymbol{\tau}[i]-1} - z_{i-1}^\star\right\|^2\right) + L\|z_{\boldsymbol{\tau}[i]} - z_i^\star\|$$

$$\leq \sum_{i=1}^{k}\frac{\mu(\boldsymbol{\tau}[i]-1)}{2}\left\|z_{\boldsymbol{\tau}[i]} - z_i^\star\right\|^2 + L\|z_{\boldsymbol{\tau}[i]} - z_i^\star\|$$

$$\tag{44}$$

It remains to bound $\|z_{\boldsymbol{\tau}[i]} - z_i^\star\|$. Let $z_{i,\boldsymbol{\tau}[i]}^\star = \arg\min_{z\in\mathcal{K}}\sum_{t=1}^{\boldsymbol{\tau}[i]} f_t^{-i}(z)$. By the definition of our unlearning algorithm,

$$\|z_{\boldsymbol{\tau}[i]} - z_i^\star\| \leq \|\xi_i\| + \|\mathcal{U}_{\text{aux}}(\tilde{z}_{\boldsymbol{\tau}[i]}, \cdot) - z_i^\star\|$$

$$\leq \|\xi_i\| + \left\|\mathcal{U}_{\text{aux}}(\tilde{z}_{\boldsymbol{\tau}[i]}, \cdot) - z_{i,\boldsymbol{\tau}[i]}^\star\right\| + \left\|z_{i,\boldsymbol{\tau}[i]}^\star - z_i^\star\right\|$$

$$\leq \|\xi_i\| + \left\|\mathcal{U}_{\text{aux}}(\tilde{z}_{\boldsymbol{\tau}[i]}, \cdot) - z_{i,\boldsymbol{\tau}[i]}^\star\right\| + 0$$

$$\leq \|\xi_i\| + \left\|\mathcal{U}_{\text{aux}}(\tilde{z}_{\boldsymbol{\tau}[i]}, \cdot) - z_{i,\boldsymbol{\tau}[i]}^\star\right\|$$

$$\tag{45}$$

where the second inequality follows by the strong growth condition. Specifically, by SGC and convexity, $z_k^\star$ is the minimizer of all remaining cost functions $f_t$ where $t \notin \boldsymbol{u}[]$. Therefore, $z_k^\star$ is the minimizer of all $f_t^{-i}(z)$ and thus the minimizer of $\sum_{t=1}^{\boldsymbol{\tau}[i]} f_t^{-i}(z)$ and $f_\ell$. That is, for all $i \in [k]$

$$z_{i,\boldsymbol{\tau}[i]}^\star = \arg\min_{z\in\mathcal{K}}\sum_{t=1}^{\boldsymbol{\tau}[i]} f_t^{-i}(z) = z_k^\star, \quad \text{and} \quad z_{i,\boldsymbol{\tau}[i]}^\star = \arg\min_{z\in\mathcal{K}} f_\ell(z).$$

Hence,

$$\left\|\mathcal{U}_{\text{aux}}(\tilde{z}_{\boldsymbol{\tau}[i]}, \cdot) - z_{i,\boldsymbol{\tau}[i]}^\star\right\| = \left\|\mathcal{U}_{\text{aux}}(\tilde{z}_{\boldsymbol{\tau}[i]}, \cdot) - \arg\min_{z\in\mathcal{K}} f_\ell(z)\right\| \overset{(a)}{\leq} \gamma^{I_2}\left\|\tilde{z}_{\boldsymbol{\tau}[i]} - \arg\min_{z\in\mathcal{K}} f_\ell(z)\right\| \leq \gamma^{I_2}D \overset{(b)}{\leq} \frac{D}{\sqrt{i^\omega}\boldsymbol{\tau}[i]L} \leq \frac{D}{\boldsymbol{\tau}[i]L},$$

$$\tag{46}$$

where step (a) follows the convergence guarantee of gradient descent (Lemma E). Step (b) is due to our lower bound on $\mathcal{I}_i$ such that $\gamma \mathcal{I}_i \leq \frac{1}{\sqrt{i^\omega}\boldsymbol{\tau}[i]L}$.

*Lemma* E (Convergence of gradient descent Wang (2020)). If the loss function $\ell$ is $\mu$-strongly convex and $\beta$-smooth, then the output $w_t$ of $T$-step gradient descent on $S$ with learning rate $\eta = \frac{2}{\mu+\beta}$ and initialization $w_0$ satisfies

$$\|w_t - w^\star\|_2 \leq \gamma^T \|w_0 - w^\star\|_2,$$

where $\gamma = \frac{\beta/\mu-1}{\beta/\mu+1}$ and $w^\star = \arg\min_w \sum_{x \in S} \ell(w,x)$ is the minimizer of the loss function $\ell$ on the dataset $S$.

Finally, we bound $\|\xi_i\|$, for some global constant $c$ independent of $T, k$ and other coefficient,

$$\mathbb{E}\left[\|\xi_i\|\right] = \sqrt{d}\sigma_i = \sqrt{\frac{d\omega i^\omega \gamma^{2I_i}L^2}{2(\omega-1)\varepsilon}} \leq \frac{1}{\boldsymbol{\tau}[i]}\sqrt{\frac{d\omega}{2(\omega-1)\varepsilon}} \tag{47}$$

where the last inequality follows by selecting $I_i \geq \frac{\omega \log i + 2\log \boldsymbol{\tau}[i] + 2\log L}{2\log\frac{1}{\gamma}}$ and thus $\gamma^{2I_i}L^2 \leq \frac{1}{\boldsymbol{\tau}[i]^2 i^\omega}$.

Substitute Equations (46) and (47) into Equation (45), we get

$$\|z_{\boldsymbol{\tau}[i]} - z_i^\star\| \leq \frac{D}{\boldsymbol{\tau}[i]L} + \frac{1}{\boldsymbol{\tau}[i]}\sqrt{\frac{d\omega}{2(\omega-1)\varepsilon}} = \frac{1}{\boldsymbol{\tau}[i]}\left(\frac{D}{L} + \sqrt{\frac{3d}{\varepsilon}}\right) \tag{48}$$

where the last equality follows by selecting $\omega = 1.2$.

Substitute Equations (45) and (47) into Equation (44),

$$\begin{aligned}
\mathbb{E}\left[\text{jump + deletion cost}\right] &\leq \sum_{i=1}^{k} \frac{\mu\boldsymbol{\tau}[i]}{2}\frac{1}{\boldsymbol{\tau}[i]^2}\left(\frac{D}{L} + \sqrt{\frac{3d}{\varepsilon}}\right)^2 + \frac{L}{\boldsymbol{\tau}[i]}\left(\frac{D}{L} + \sqrt{\frac{3d}{\varepsilon}}\right) \\
&= \sum_{i=1}^{k} \frac{\mu}{2\boldsymbol{\tau}[i]}\left(\frac{D}{L} + \sqrt{\frac{3d}{\varepsilon}}\right)^2 + \frac{L}{\boldsymbol{\tau}[i]}\left(\frac{D}{L} + \sqrt{\frac{3d}{\varepsilon}}\right) \leq \frac{C'dD^2L^2}{\min(\varepsilon,\sqrt{\varepsilon})}\sum_{i=1}^{k}\frac{1}{\boldsymbol{\tau}[i]}
\end{aligned} \tag{49}$$

for a global constant $C'$ independent of all $L, \beta, \mu, d, k, T$.

Combining equation Equations (43) and (49), we get

$$\mathbb{E}\left[\sum_{i=0}^{k}\sum_{t=\boldsymbol{\tau}[i]+1}^{\boldsymbol{\tau}[i+1]}\left(f_t(z_t) - f_t(z_i^\star)\right)\right] \leq \frac{CL^2\log T + kD}{2\mu} + \frac{C'dD^2L^2}{\min(\varepsilon,\sqrt{\varepsilon})}\sum_{i=1}^{k}\frac{1}{\boldsymbol{\tau}[i]} \tag{50}$$

This concludes the proof. □

# D Experimental Details

Below, we present the experimental details on comparing passive OLU, active OLU, and online DP in Section 5.

**Loss model and comparator.** The horizon is $T = 1000$, the dimension is $d = 10$, and outputs are projected onto the Euclidean ball $\{w : \|w\|_2 \leq 1\}$. Each example is a normalized Gaussian feature vector $x_t \in \mathbb{R}^{10}$ with $\|x_t\|_2 = 1$. For the reported run the ground truth is stationary: a unit vector $w^\star$ is sampled once per seed and $y_t = x_t^\top w^\star$. The per-round loss is

$$f_t(w) = \frac{1}{2}(x_t^\top w - y_t)^2 + \frac{\gamma_{\text{reg}}}{2}\|w\|_2^2,$$

with $\gamma_{\text{reg}} = 1$, strong-convexity parameter $\mu = 1$, smoothness upper bound $\beta = 2$, and gradient bound $L = 3$. Dynamic regret is computed against a piecewise-constant retained-data comparator: after each deletion, the comparator is recomputed by solving the constrained ridge problem on all examples except those deleted so far.

**Deletion protocol.** Each run has $k = 10$ deletions. Given deletion times $\tau_i$, the deleted index $u_i$ is drawn uniformly without replacement from the current block $\{\tau_{i-1} + 1, \ldots, \tau_i\}$, with $\tau_0 = 0$. The three schedules are

$$
\begin{aligned}
&\texttt{start:} && 3, 4, 5, 6, 7, 8, 9, 10, 11, 12, \\
&\texttt{uniform:} && 3, 114, 224, 335, 446, 556, 667, 778, 888, 999, \\
&\texttt{end:} && 990, 991, 992, 993, 994, 995, 996, 997, 998, 999.
\end{aligned}
$$

The "start" schedule simulates a harder setting for passive OLU and the "end" schedule simulates the easier setting for the passive OLU.

**Algorithms.** The passive OLU method runs projected OGD with step size $\eta_t = 1/(\mu t)$ as described in Algorithm 1. At deletion $i$, it adds Gaussian deletion noise $\mathcal{N}(0, \sigma_i^2 I_d)$ and projects back to the ball, where

$$
\sigma_i = \frac{L}{\mu \tau_i} \sqrt{\frac{3 i^{1.2}}{\varepsilon_{\mathrm{OLU}}}}, \qquad \varepsilon_{\mathrm{OLU}} = 7.
$$

The active OLU method uses the same OGD updates and then performs active unlearning procedure described in Algorithm 2. At each deletion it performs $I = 20$ projected descent-to-delete steps with step size $2/(\beta + \mu)$ on the first retained example in the prefix, then adds calibrated Gaussian noise. With $\omega = 1.2$, its deletion noise variance is

$$
\sigma_i^2 = \frac{\omega i^\omega \gamma^{2I} L^2}{2(\omega - 1)\varepsilon_{\mathrm{OLU}}}, \qquad \gamma = \frac{\beta - \mu}{\beta + \mu}.
$$

The online-DP baseline is Gaussian PFTAL with clipped gradients (Algorithm 1 in Guha Thakurta and Smith (2013)). Gradients are clipped to the true gradient upper bound $L = 3$ and accumulated with a binary tree aggregator of depth $\lceil \log_2 T \rceil = 10$. The total privacy parameters are $\varepsilon_{\mathrm{DP}} = 21$ and $\delta_{\mathrm{DP}} = 10^{-3}$, split uniformly across tree levels, giving per-node $\varepsilon = 2.1$, $\delta = 10^{-4}$, and Gaussian noise scale $6.2051604341$. The privacy parameters $(\varepsilon_{DP}, \delta_{DP})$ was selected by applying the RDP-to-approximate-DP conversion to ensure a fair comparison with other algorithms that satisfy $(\alpha, \alpha \varepsilon_{OLU})$-Rényi OLU. This DP baseline protects every update, whereas the OLU algorithms protect deleted points, so the experiment should be read as a regret comparison under different privacy notions.

