# OpenReview forum: "Online Learning and Unlearning: Efficient Algorithms with Near-Optimal Regret Guarantees"
_TMLR — Under review for TMLR_

### Review · Reviewer_q9v6 · 2026-05-04

**Summary Of Contributions:**

The paper studies online learning-unlearning (OLU)  in the online convex optimization setting, where a learner updates a model sequentially on a stream of convex losses while accommodating occasional unlearning requests between updates. The paper proposes two OLU algorithms: passive OLU and active OLU. Under standard convexity and smoothness conditions, the proposed methods achieve regret bounds comparable to standard online gradient descent. This shows that strong online unlearning guarantees can be achieved with minimal loss in learning performance.

**Audience:**

Yes

**Audience Explanation:**

I believe so. Unlearning is a popular topic in recent machine learning literature. It is related to privacy risks and regulations that should be of interest to many readers of TMLR.

**Claims And Evidence:**

Yes

**Claims Explanation:**

I believe so. The paper provides rigorous definitions and the detailed proofs in the appendix, although I believe the writings can be improved, and some notations can be made more clear.

**Requested Changes:**

(1) On page 1, perhaps there is no need to introduce the mathematical symbol $\tau$, unless you provide more descriptions about it. Later, on page 3, you have $\boldsymbol{\tau}$. I do not know whether they are the same or not. Also, on page 2, when $T$ first appears, please define its meaning (even though it might be obvious to most readers.)

(2) In the caption of Table 1, what is the little $t$ in $O(\log t)$?

(3) Right after equation (1), please define $f_{1:t-1}$ and $z_{1:t-1}$.

(4) In equation (1), $g_{t}(f_{1:t-1},z_{1:t-1}).$ should be $g_{t}(f_{1:t-1},z_{1:t-1}),$

(5) In equation (2), $(z_{t-1})].$ should be $(z_{t-1})],$

(6) After equation (2), you should mention that $\eta_{t}>0$ is the learning rate.

(7) In equation (3), $g_{t}.$ should be $g_{t},$

(8) In the paragraph after equation (3), you wrote $z_{t+1}=z_{t}-\eta\nabla f_{t}(z_{t})$. But in equation (2), the stepsize $\eta_{t}$ is time-dependent and you also have a projection operator. Also, in Algorithm 1, for OGD, it seems you do not have the projection operator but you have that in equation (2). That is a bit confusing.

(9) In the last paragraph in Section 2, you started discussing $(\varepsilon,\delta)$-DP and $(\alpha,\varepsilon)$-RDP. I think it would be a good idea to provide formal definitions of these two concepts in case the readers are not familiar with differential privacy literature.

(10) It would be great if more discussions on condition C3 in Section 3.1. can be added. For example, is this condition common in the literature? If not, is it similar to some conditions available in the literature? Also, do you have some popular examples that can satisfy C1, C2 and C3 simultaneously?

(11) In Corollary 3.2., you take $\gamma=1$ for the convex case and $\gamma_t=1-\frac{1}{t}$ for the strongly convex case. But in C2, $\gamma$ is constant. Did you want to use $\gamma_t$ in C2 instead to be more general?

(12) In the beginning on page 8, please define $\tilde{O}$.

(13) It is great that you have some discussions on the optimality of the regret bound dependence on $T$ after Theorem 3.4. Is it possible you also add some discussions on the dependence of the bound on $T$ about Theorem 3.3. and Theorem 4.2.?

(14) In reference Altschuler and Talwar, langevin should be Langevin.

---

> ### Author Response · Authors · 2026-05-26
>
> We thank the reviewer for their careful reading and constructive feedback. In response, we have revised the manuscript to improve the clarity, highlighting the updated text in yellow. We outline these modifications and provide individual responses to the questions below.
>
> **Notation for $\tau$, $T$, and related symbols.**  We thank the reviewer for pointing this out. We have revised the text to define $\tau$ as the deletion time and $T$ as the time horizon when they first appear.
>
> **Meaning of $t$ in $O(\log t)$ in Table 1.**  We have clarified the caption of Table 1. The notation $O(\log t)$ refers to the computational cost at each time step $t$, in contrast to our unlearning algorithms, which incur no additional cost on non-deletion rounds.
>
> **Definitions of $f_{1:t-1}$, $z_{1:t-1}$, and $\eta_t$.**  We thank the reviewer for noting these omissions. We now define $f_{1:t-1}=\{f_1,\ldots,f_{t-1}\}$,
>    $z_{1:t-1}=\{z_1,\ldots,z_{t-1}\},$  immediately after Equation (1). We have also clarified after Equation (2) that $\eta_t$ denotes the learning rate at time $t$.
>
> **Punctuation in Equations (1), (2), and (3), and typos in the reference.**  Thank you for catching these typographical issues. We have corrected the punctuation in Equations (1-3), and the typo in the bibiography.
>
> **Consistency of the projection steps in OGD between Equation (2), the paragraph after Equation (3), and Algorithm 1.**  We agree that the previous version was confusing. Equation (2) uses a time-dependent stepsize and a projection operator, but the surrounding text and Algorithm 1 were not fully aligned. We have revised the paragraph after Equation (3) and updated the OGD version of Algorithm 1 so that both consistently include the time-dependent learning rate and projection step.
>
> **Definitions of DP and RDP.**  We thank the reviewer for this suggestion. We have added formal definitions of $(\varepsilon,\delta)$-differential privacy and $(\alpha,\varepsilon)$-Rényi differential privacy in Appendix A.1 and a detailed discussion on the conversion from RDP to DP guarantee.
>
> **Discussion of Condition C3.**  We have expanded the discussion of Condition C3 in Section 3.1. For OGD, it is implied by standard assumptions. For example, if the loss functions are $L$-Lipschitz, then  $\|g_t(f,z)-z\|_2 \le \eta_t L$, so C3 holds with $\Delta_t=\eta_t L$. Thus, for OGD, C3 is weaker than Lipschitzness. We have also added examples showing that C1, C2, and C3 are simultaneously satisfied by projected OGD with convex, smooth, and Lipschitz losses, as well as by projected OGD with strongly convex, smooth, and Lipschitz losses.
>
> **Time-varying $\gamma_t$ in Condition C2.**  Thank you for pointing this out. We have revised Condition C2 to allow a time-dependent contraction coefficient $\gamma_t$.
>
> **Definition of $\tilde O$.**  We have added a definition of $\tilde O$ at its first appearance. Specifically, $\tilde O(\cdot)$ denotes big-O notation with logarithmic factors omitted; that is, $f(x)=\tilde O(g(x))$ if $f(x)\le Cg(x)\operatorname{polylog}(x)$ for some constant $C<\infty$.
>
> **Dependence on $T$ in Theorems 3.3 and 4.2.**  We thank the reviewer for this suggestion. We have added discussion before Theorems 3.3 and 4.2 on the dependence of the regret bounds on $T$. In both results, under favorable deletion schedules, the regret has an $O(\log T)$ dependence for strongly convex losses, matching the standard rate of online learning without unlearning. This shows that, in these settings, the proposed unlearning algorithms can be optimal with respect to the time horizon $T$, while the additional terms capture the cost of handling deletions.

---

> > ### Comment · Reviewer_q9v6 · 2026-05-31
> > **response**
> >
> > Thanks for addressing the points I raised!

---

### Review · Reviewer_5HCd · 2026-05-10

**Summary Of Contributions:**

The submission introduces the Online Learning-Unlearning (OLU) framework, which formalizes how a learner can satisfy deletion requests in a streaming setting while maintaining low regret. Two algorithms are proposed---Passive OLU and Active OLU---both built on Online Gradient Descent (OGD), with certified $(\alpha, \varepsilon)$-unlearning guarantees via Renyi divergence.

**Audience:**

Yes

**Audience Explanation:**

This submission has interesting novelty and motivation. The problem formulation is genuinely new and well-motivated. Extending machine unlearning to the online/streaming setting is a natural and important gap, and the paper is the first to address it formally. The comparison to the turnstile model in continual observation is a useful anchoring point.

**Broader Impact Concerns:**

There is no impact concerns for this submission.

**Claims And Evidence:**

Yes

**Claims Explanation:**

The authors have presented all the mathematical analysis and proofs in the appendix.

**Requested Changes:**

The regret definition (Equation~5) needs more justification. The OLU regret uses a different comparator $z_i^\star$ for each interval between deletions. This is analogous to dynamic regret, but the paper does not discuss whether this makes the bound harder or easier to achieve compared to static regret. Since $z_i^\star$ changes with deletions, it is possible an adversary could engineer deletions to make the comparator sequence favorable. The relationship to standard static regret benchmarks should be discussed more explicitly.

Assumption 4.1 (Strong Growth Condition) is restrictive and underexplored. The SGC limits the adversary's power significantly---the paper acknowledges this but does not quantify how restrictive it is in practice, or give natural examples where it holds in an online adversarial setting. Since SGC is central to Active OLU's guarantees, this deserves more discussion or at least a worked example.

The convex case regret bound has an unresolved extra $\sqrt{\log T}$ factor. The paper candidly notes this gap (Section~3.2) and attributes it to the regularization technique, but does not resolve it. This is a meaningful gap: $O(\sqrt{T \log T})$ vs.\ $O(\sqrt{T})$ is non-trivial for large $T$. A lower bound argument, or even a conjecture about whether this is avoidable, would strengthen the paper.

The schedule-robustness mechanism (Algorithm~3 / Corollary~3.6) is somewhat under specified. The corollary bounds regret in terms of $\ell$ (the number of restarts), but $\ell$ itself depends on the deletion schedule and the threshold criterion. The paper does not characterize the worst-case $\ell$ under adversarial schedules, so the bound's practical meaning is unclear. A concrete example of a bad schedule and the resulting $\ell$ would help.

There are no numerical experiments in this submission. For a paper targeting TMLR, the complete absence of empirical validation is a significant weakness. Even simple synthetic experiments---e.g., comparing regret and noise magnitude across methods on a quadratic loss with varying deletion schedules---would substantially strengthen the practical claims. The theoretical guarantees are meaningful, but TMLR reviewers will likely expect some empirical grounding.

Proposition~1 (DP baseline comparison) is somewhat informal. Translating $(\varepsilon, \delta)$-DP to $(\alpha, \varepsilon)$-RDP and then to OLU requires careful accounting, and the paper uses phrasing like ``interpreting these as \ldots\ yields'' without a complete derivation in the main text. This is a critical baseline comparison and deserves a tighter treatment.

There are also the following minor issues:

The notation in Table~1 uses undefined symbols (e.g., the ``C'' subscript for convex vs.\ ``SC'' for strongly convex) without a legend directly in the table caption.

The proof sketch for Theorem 3.1 is helpful, but Figure 1's description of ``subsequent deletions is more involved'' is left vague---a brief explanation of the multi-deletion case in the sketch would improve readability.

The condition in Corollary~3.2 states $\eta \leq 2/\beta$ gives $\gamma = 1$ for convex losses, but $\gamma = 1$ means no contraction at all, which seems to undermine the noise calibration argument. This should be clarified---does the algorithm still work in this degenerate case?

---

> ### Author Response · Authors · 2026-05-26
>
> We thank the reviewer for their careful reading and constructive feedback. In response, we have added numerical experiments to complement our theoretical findings and revised the manuscript for clarity, highlighting the updated text in yellow. We outline these modifications and provide individual responses to the questions below.
>
> **Numerical experiments**
> We thank the reviewer for this constructive suggestion. We have added numerical simulations to empirically compare the methods presented in Table 2. We also consider the effect of different deletion schedules: an easier schedule where the deletion requests occur near the end of training, a moderate schedule where these requests are uniformly distributed, and an adversarial schedule where the deletions happen at the start. The empirical results consistently align with the theoretical upper bounds shown in the table: active and passive OLU consistently outperform online DP in all three deletion schedules, and their efficiency depends heavily on the deletion schedule. We have added a new paragraph in Section 5 to discuss these findings, and the implementation details are now included in Appendix D.
>
> **Regret definition needs more justification**
> We thank the reviewer for raising this interesting point. We agree that Equation 5 deserves more discussion, and we added a paragraph following Equation 5 to clarify this point. Specifically, the comparator in Equation 5 is not chosen to make the benchmark favorable to the learner. Rather, it is dictated by the unlearning requirement: after the $i$-th deletion, the learner is required to behave like an algorithm retrained on the sequence with the first $i$ deleted cost functions removed. Therefore, the natural regret benchmark during the interval between the $i$-th and $(i+1)$-st deletion is the best-in-hindsight solution for that same post-deletion dataset. This ensures that the learner and comparator are evaluated with respect to the same retained history.
>
> The OLU regret is not uniformly easier or harder than standard static regret. Let $z^\star$ denote a fixed static comparator. Then, the difference between the OLU regret and static regret can be written as $R_{\mathrm{OLU}} - R_{\mathrm{static}} = \sum_{i=0}^k \sum_{t=\tau[i]+1}^{\tau[i+1]} \left(f_t(z^\star)-f_t(z_i^\star)\right)$. This term can be either positive or negative, depending on the loss sequence and deletion requests. Thus, adversarially chosen deletions can in principle make the benchmark either more or less favorable. However, our results do not rely on assuming that deletion requests are benign or independent of the online-learning adversary.
>
> **The comparison with online DP is informal**
> We thank the reviewer for pointing this out.We thank the reviewer for pointing this out. We have revised the discussion to avoid conflating Rényi-style and approximate-DP-style indistinguishability. For a fair comparison with online DP baselines, we now convert our Rényi-style OLU guarantee to approximate OLU before comparing regret bounds. We also include the formal definition and conversion proof in Appendix A.1.
>
> Importantly, for a large range of $\epsilon, \delta$, the comparison shown in the table will not be affected, as discussed below. As both passive and active OLU satisfy $(\alpha, \alpha\varepsilon_R)$-OLU, a stronger notion than approximate-DP style indistinguishability guarantee. Applying the standard RDP-to-approximate-DP conversion (by proposition 3 in [1]), we can convert our $(\alpha, \alpha \varepsilon_R)$-Renyi OLU guarantee to $(\varepsilon, \delta)$-approximate OLU guarantee with $\varepsilon = 3\varepsilon_{\mathrm{R}}.$ Therefore, the asymptotic comparisons in the table remain unchanged up to constant factors, for $\log 1/\delta = \Theta (\varepsilon)$.
>
> **Schedule-robust algorithm still depends on $\ell$ and in the worst case, and add a concrete example of a worst case $\ell$.**
> We thank the reviewer for proposing this change. We added a paragraph clarifying this point in the paragraph below proposition 3.6. The bound depends on the deletion schedule through $\ell$, which measures how many times the schedule violates the criterion and triggers a restart. In the worst case, $\ell$ can be as large as $k$ if every deletion results in a restart. Nevertheless, the restart mechanism prevents unfavorable schedules from causing uncontrolled degradation in regret, since the algorithm resets instead of continuing to accumulate excessively large noise.
>
> [1] Ilya Mironov. Rényi differential privacy. 2017 IEEE 30th Computer Security Foundations Symposium (CSF). 2017.

---

> ### Author Response · Authors · 2026-05-26
>
> **Assumption 4.1 SGC limits the adversary's power**
> We agree that Assumption 4.1 restricts the class of admissible loss sequences for Active OLU. This assumption is needed because the offline unlearning subroutine (the descent-to-delete algorithm) used in Active OLU is designed for ERM, whereas the online trajectory is generated by OGD. The Strong Growth Condition allows us to control the discrepancy between these two trajectories.
>
> A simple setting where the Strong Growth Condition holds is a family of quadratic losses sharing a common minimizer $w^\star$, namely $f_t(w)=\frac{a_t}{2}\|w-w^\star\|^2$, where $a_t>0$. For any retained set $R_j$ after $j$ deletions, SGC holds with $B_j=(\max_{t\in R_j}a_t)/\sum_{t\in R_j}a_t$. This example captures a realizable setting in which the retained losses are aligned around a shared solution. Our numerical experiments are conducted in this common-minimizer setting.
>
> We will also clarify the limitation of the assumption. SGC can fail when retained losses exhibit strong gradient cancellation. For example, if $f_1(w)=\frac{1}{2}(w-1)^2$ and $f_2(w)=\frac{1}{2}(w+1)^2$, then at $w=0$ the aggregate gradient is $\nabla F(0)=0$, while the individual gradients are nonzero. Hence no finite constant $B$ can satisfy SGC. This illustrates that Active OLU excludes fully adversarial retained loss sequences, even though heterogeneous or corrupted losses may still appear in the stream if they are deleted before SGC is required to hold.
>
>  **Lower bound argument / conjecture to strength the convex case towards resolving the additional $\sqrt{\log T}$ factor**
>
> We agree that the extra $\sqrt{\log T}$-type factor in the convex case is a meaningful gap and should be discussed more explicitly. This factor arises from the regularization step in Theorem 3.4. To apply the strongly convex passive-unlearning analysis to general convex losses, we replace each loss by the surrogate $\widetilde f_t(z)=f_t(z)+\frac{\lambda}{2}\|z\|^2$. This introduces a tradeoff between the strongly convex regret term, which scales as $\frac{\log T+2k}{\lambda}$, and the regularization bias, which scales as $TD^2\lambda$. Balancing these two terms gives $\lambda \asymp \sqrt{\frac{\log T+2k}{T}}$, leading to the dependence $\sqrt{T(\log T+2k)}$.
>
> Thus, the additional $\sqrt{\log T}$ factor is a consequence of inducing curvature through regularization. We will revise the discussion to make this source of the gap clearer and to emphasize that removing it remains an open question. As noted in Section 5, obtaining meaningful lower bounds for unlearning is challenging even in the offline settings, and understanding whether the $\sqrt{\log T}$ gap is inherent in online unlearning or merely an artifact of our regularization-based analysis is an important direction for future work. We will also clarify that Theorem 3.5 partially mitigates this issue under additional assumptions and access to public gradient-norm information, but does not resolve the worst-case purely convex setting.
>
> **Clarification on $\gamma$ for corollary 3.2**
> Indeed, the approach remains valid for $\gamma = 1$ as the trajectory is non-expansive. Notably, our regret bound for the convex setting consistently assumes this upper bound of $\gamma = 1$ throughout the entire analysis.
>
> **Improving the clarity of the legend in Table 1 and other notational issues** We have implemented the reviewer's suggestions to improve the legend in Table 1. Additionally, we have addressed all minor notational inconsistencies to enhance the overall readability of the manuscript.

---

### Review · Reviewer_LnZD · 2026-05-18

**Summary Of Contributions:**

This paper derives regret bounds for active and passive (un)learning algorithms in the formalism of online learning. The main results are unlearning guarantees and regret bounds for (i) passive unlearning in the convex and strongly convex case, and (ii) for active unlearning in the strongly convex case together with the strong growth condition.

**Strengths:**
The main appeal of these results is that they allow to compare different (un)learning strategies in terms of their computational cost, as well as their regret, in a precise and formal theoretical framework (see Table 1 in the paper). Besides the theoretical contributions, the paper is very well structured and can be followed well despite the technical nature of the results.

**Weaknesses:**
One evidently missing piece is the empirical picture for the presented (un)learning algorithms; it would be interesting to see how the algorithms from Table 1 compare in such a setup, even for a synthetical/toy data example. Unfortunately, the paper does not make any efforts in that direction. Given that regret bounds might be not tight, an empirical comparison would add additional value to the submission in my view.

**Questions:**

- For passive unlearning in the convex case, it seems that the worst-case schedule is examples that appear early in training, and where the deletion request appears late ($u[i]=1, \tau[i] \approx T$). It would be interesting to see whether this is an artefact of the proof, or whether this can be observed empirically.
- Related to the above, Table 1 assigns the cost of retraining to be $\tau[i]$. However, it seems that the actual cost is related to the differences $\tau[i] - u[i]$, as for example if we have a single deletion request at $\tau[i]=T$ and $u[i]=T-1$, then the cost of retraining in this specific case is $1$ and not $T$. Maybe the authors can clarify this during the discussion period.
- The current submission does not discuss related work extensively. For readers that are less familiar with previous results on unlearning algorithms, and with techniques used in practice, it would set the context to have a short summary of the most related literature.
- Regarding the results with restarts (Corollary 3.6): maybe I misunderstood something here, but if we have a single deletion request with $\tau=T$ and $u=1$, then under suitable choice of T we can always trigger the restart condition. In this case, the algorithm will almost surely end at a random point, as it was restarted in the last step. So how can we obtain a regret bound in this case?


**Minor remarks/questions:**

- Maybe the authors can add a definition of $\alpha$-Renyi divergence.
- Algorithm 3, Line 3 has a different font than the rest.
- Algorithm 2, Line 9 is unclear: which remaining set, and what does $\ell$ index? Can the remaining set be empty?
- Page 4, equation in the middle: it is written once $u[1:i]$ and once $u[:i]$, do these two notations mean the same?
- For the intuition of Theorem 3.1, it might be also insightful to write the current iterate as a weighted average of past gradients (in the case of gradient descent), and see how removing/unlearning a single gradient can be undistinguishable from noise.
- page 8: "$\mathcal{G}_1(\tau, u)$ is constant". Constant in which quantities?

**Audience:**

Yes

**Audience Explanation:**

Algorithms for machine unlearning are highly relevant in practice, and hence their theoretical understanding is interesting.

**Claims And Evidence:**

Yes

**Claims Explanation:**

The theoretical results are supported by accurate explanations and proofs.

**Requested Changes:**

- In Table 1, the dimensional dependence is not applied consistently for all methods: for the active and passive methods, their regret bounds have also some dependence on $d$, but this is sometimes omitted in the table.
- Adding a small experimental comparison to see whether some of the results are theroetical/wors-case artefacts would be a nice improvement.
- Please also see the Questions sectiona above.

---

> ### Author Response · Authors · 2026-05-26
>
> We thank the reviewer for their careful reading and constructive feedback. In response, we have added numerical experiments to complement our theoretical findings and revised the manuscript for clarity, highlighting the updated text in yellow. We outline these modifications and provide individual responses to the questions below.
>
> ## Part I Summary of Revisions
> **Additional empirical justifications**
> We have added numerical simulations to empirically compare the methods presented in Table 2. Specifically, we consider strongly convex and smooth quadratic loss functions, defined as $f_t(w) = \frac{1}{2}(x_t^\top w - y_t)^2 + \frac{1}{2}\|w\|_2^2$. The targets are generated as $y_t = x_t^\top w^* $ using randomly sampled inputs $x_t$ and a fixed true parameter $w^*$ drawn from the unit sphere. Then, we investigate how deletion schedules impact performance across three scenarios: (1) an adversarial setting for passive OLU featuring initial deletions, (2) a moderate setting with uniformly distributed deletions, and (3) a favorable setting with terminal deletions. Consistent with the comparison of theoretical regret upper bounds shown in Table 2, our empirical analysis on the simulated example shows that active and passive OLU consistently outperform online DP in all three scenarios, and that their efficiency depends heavily on the deletion schedule. We have added a new paragraph in Section 5 to discuss these findings, and the implementation details are now included in Appendix D.
>
> **Discuss related work**
> We thank the reviewer for this valuable suggestion. We have added a paragraph to the Preliminaries section discussing several existing offline certified unlearning algorithms to better contextualize our work. In particular, we discuss certified unlearning methods for convex ERM that rely on first-order information from the retained dataset [1] or second-order information associated with the deleted samples [2]. We also discuss approaches based on noisy gradient descent over the retained dataset, which provide certified unlearning guarantees beyond convex ERM, including settings with non-convex losses [3,4].
>
>
> **Inconsistency with respect to the dimension $d$ i n Table 1 and minor remarks with notation**
> We thank the reviewer for their careful reading and for pointing out the inconsistency regarding the dimension dependence in the table. We have updated the table to include the dependence on $d$ as suggested.
>
> **Algorithm 2, Line 9**
> We thank the reviewer for pointing out the overloaded notation $\ell$, which we have corrected in the updated draft. In this context, $\ell$ refers to the index of the first function within the set of cost functions that have not been deleted up to time $\tau[i]$. Because the total number of deletions $k$ is always strictly less than the number of time steps $t$ (meaning there are always some remaining cost functions), this set cannot be empty.
>
> **Page 8 clarification**
> We have revised the manuscript to explicitly clarify that $\mathcal{G}(\tau, u)$ is constant with respect to $k$ and $T$.
>
> [1] Seth Neel, Aaron Roth, and Saeed Sharifi-Malvajerdi. Descent-to-Delete: Gradient-Based Methods for Machine Unlearning. ALT, 2021.
>
> [2] Ayush Sekhari, Jayadev Acharya, Gautam Kamath, and Ananda Theertha Suresh. Remember What You Want to Forget: Algorithms for Machine Unlearning. NeurIPS, 2021.
>
> [3] Eli Chien, Haoyu Wang, Ziang Chen, and Pan Li. Langevin Unlearning: A New Perspective of Noisy Gradient Descent for Machine Unlearning. NeurIPS, 2024.
>
> [4] Anastasia Koloskova, Youssef Allouah, Animesh Jha, Rachid Guerraoui, and Sanmi Koyejo. Certified Unlearning for Neural Networks. ICML, 2025.

---

> ### Author Response · Authors · 2026-05-26
>
> ## Part II Response to the questions
>
> **Whether early-point/late-request schedules are a proof artifact**
>
> We thank the reviewer for this insightful question. We agree that the exact form of the schedule dependence in the convex passive-OLU bound is tied to the learning-rate schedule used in the theorem. Specifically, when a point observed at time $u$ is deleted, the counterfactual trajectory first differs from the original one by a perturbation whose size is proportional to the step size $\eta_u L$ where $L$ is the Lipschitz coefficient of the loss function. In the merely convex case, subsequent OGD updates are non-expansive but not contractive, so this perturbation is not guaranteed to decay before the deletion request arrives at time $\tau$. Moreover, in the regret analysis, a state perturbation or injected noise at time $\tau$ is charged through the OGD potential term at scale $1/\eta_\tau$ (e.g. see equation (13) and (14)). Therefore, under the standard decreasing choice $\eta_t \propto 1/\sqrt{t}$, early deletions requested late are naturally the hardest case: early points can create larger perturbations because $\eta_u$ is larger, and late requests are more costly because $1/\eta_\tau$ is larger.
>
> Thus, the early-point/late-request behavior is not simply an artifact of a loose inequality in the proof for the passive OGD algorithm and step-size schedule we analyze; it reflects the absence of contraction in the general convex setting. At the same time, we agree that this should not be interpreted as a universal lower bound over all possible learning-rate schedules. A constant, horizon-dependent, or adaptive learning rate can redistribute sensitivity across time. Our regret analysis with adaptive learning rate (Theorem 3.5) reflects this effect.
>
> **Related to the above, Table 1 assigns the cost of retraining to be $\tau[i]$. However, it seems that the actual cost is related to the differences $\tau[i] - u[i]$, as for example if we have a single deletion request at $\tau[i] = T$ and $u[i] = T-1$, then the cost of retraining in this specific case is 1 and not T. Maybe the authors can clarify this during the discussion period.**
>
> We thank the reviewer for raising this important question. In Table 1, the reported cost for retraining corresponds to a retraining-from-scratch baseline without checkpointing. After receiving a deletion request at time $\tau[i]$, this baseline restarts from the initial model and replays the retained examples in the stream up to time $\tau[i]$, giving cost $O(\tau[i])$.
>
> The reviewer is correct that with additional stored state, the cost can be smaller. For example, if all intermediate iterates are checkpointed, then for a single deletion at $u[i]$ requested at $\tau[i]$ one can roll back to the checkpoint just before $u[i]$ and replay only $O(\tau[i] - u[i])$ updates. More generally, sparse or full checkpointing gives a memory-computation tradeoff: more saved states reduce retraining cost, but require additional memory not reflected in the table.
>
>
> **Question on passive OLU with restart**: We thank the reviewer for raising this edge case. The key point is that a restart resets only the internal state used for future predictions; it does not retroactively change the outputs already produced or the losses already incurred. In the example $u[i]=1$ and $\tau[i]=T$, all regret contributions up to time $T-1$ are already fixed. If the restart affects the prediction at the final time $T$, then it contributes only a single-round excess loss. Since the domain is bounded and the losses are Lipschitz, this is $O(1)$, which is absorbed by the corollary’s $O(\sqrt{\ell T(\log T+2k)})$-type bound.

---

> > ### Comment · Reviewer_LnZD · 2026-06-23
> > **Thank you!**
> >
> > I would like to thank the authors for their thoughtful responses, and the changes in the submission based on the reviews.